# Retinal input integration in excitatory and inhibitory neurons in the mouse superior colliculus in vivo

Carolin Gehr[1,2,3,4], Jeremie Sibille[1,2,3,4], Jens Kremkow[1,2,3,4]*

[1]Neuroscience Research Center, Charité-Universitätsmedizin Berlin, Berlin, Germany; [2]Bernstein Center for Computational Neuroscience Berlin, Berlin, Germany; [3]Institute for Theoretical Biology, Humboldt-Universität zu Berlin, Berlin, Germany; [4]Einstein Center for Neurosciences Berlin, Berlin, Germany

**Abstract** The superior colliculus (SC) is a midbrain structure that receives inputs from retinal ganglion cells (RGCs). The SC contains one of the highest densities of inhibitory neurons in the brain but whether excitatory and inhibitory SC neurons differentially integrate retinal activity in vivo is still largely unknown. We recently established a recording approach to measure the activity of RGCs simultaneously with their postsynaptic SC targets in vivo, to study how SC neurons integrate RGC activity. Here, we employ this method to investigate the functional properties that govern retinocollicular signaling in a cell type-specific manner by identifying GABAergic SC neurons using optotagging in VGAT-ChR2 mice. Our results demonstrate that both excitatory and inhibitory SC neurons receive comparably strong RGC inputs and similar wiring rules apply for RGCs innervation of both SC cell types, unlike the cell type-specific connectivity in the thalamocortical system. Moreover, retinal activity contributed more to the spiking activity of postsynaptic excitatory compared to inhibitory SC neurons. This study deepens our understanding of cell type-specific retinocollicular functional connectivity and emphasizes that the two major brain areas for visual processing, the visual cortex and the SC, differently integrate sensory afferent inputs.

*For correspondence:
jens.kremkow@charite.de

**Competing interest:** The authors declare that no competing interests exist.

## eLife assessment

This **valuable** study contributes to understanding how retinal activity shapes the response properties of excitatory and inhibitory neurons in a major visual target, the superior colliculus. The evidence supporting the claim is **convincing**: the work is technically excellent yet the interpretation of these results assumes an unbiased sampling and integration of the RGC axon in the SC, a caveat pointed out by the authors. Overall, this study provides insights into the integration of visual information from the eye to the brain, and this work will be of interest to visual neuroscientists.

## Introduction

The mouse superior colliculus (SC) is a midbrain structure that receives direct inputs from retinal ganglion cells (RGCs) (*Dräger and Hubel, 1976*; *Ellis et al., 2016*; *Huberman et al., 2008*; *Kremkow and Alonso, 2018*). Together with the visual cortex (*Glickfeld et al., 2013*; *Lashley, 1931*; *Niell and Scanziani, 2021*; *Petruno et al., 2013*), the SC plays a key role in visually guided behaviors (*Evans et al., 2018*; *Hoy et al., 2019*; *Shang et al., 2019*; *Shang et al., 2015*; *Wei et al., 2015*). Intriguingly, the visual SC layers contain one of the highest densities of inhibitory (GABAergic) neurons in the brain (*Kaneda et al., 2008*; *Mize, 1992*) suggesting that inhibition plays a key role in visual processing within the SC. Indeed, SC inhibitory neurons (INs) are known to be involved in several sensory functions

including surround suppression (*Kasai and Isa, 2016*) and motion processing (*Barchini et al., 2018*; *Gale and Murphy, 2016*), but also in the regulation of wakefulness (*Zhang et al., 2019*). However, how GABAergic SC neurons are recruited by their retinal inputs in vivo remains largely unknown (*Shi et al., 2017*). An understanding of how RGC inputs are integrated by SC INs and SC excitatory neurons (EXNs) is crucial for reaching a mechanistic understanding of the computations within the SC microcircuit.

In sensory systems, the divergence of long-range afferent axons onto GABAergic and non-GABAergic neurons ensures a balance between excitation and inhibition (*Miller, 2016*). For example, in sensory cortices, thalamic afferents differentially activate EXNs and local INs (*Cruikshank et al., 2007*; *Ji et al., 2016*) with a stronger drive onto INs compared to EXNs (*Bruno and Simons, 2002*; *Cruikshank et al., 2007*; *Gabernet et al., 2005*; *Gibson et al., 1999*; *Inoue and Imoto, 2006*; *Swadlow, 2003*). This stronger drive on INs establishes an effective feedforward inhibition (*Agmon and Connors, 1992*; *Bereshpolova et al., 2020*; *Gabernet et al., 2005*; *Kremkow et al., 2010a*; *Kremkow et al., 2010b*; *Sun et al., 2006*) that balances the excitation from thalamic afferents (*Isaacson and Scanziani, 2011*) and contributes to sharpening stimulus feature selectivity (*Lee et al., 2012*). Moreover, this wiring motif can partly be linked to the response properties of cortical INs as they typically show higher firing rates, broader tuning, low selectivity, and high sensitivity (*Alonso and Swadlow, 2005*; *Bruno and Simons, 2002*; *Gibson et al., 1999*; *Kremkow et al., 2016*; *Porter et al., 2001*; *Swadlow et al., 2002*) to facilitate feedforward inhibition in response to thalamic input. Despite the importance of this afferent wiring motif for sensory processing in the thalamocortical visual circuit, it is currently unknown whether the retinocollicular pathway follows similar or different principles.

We recently demonstrated that SC neurons can receive strong input from RGC axons in vivo (*Sibille et al., 2022a*). Moreover, around one third of retinocollicular synapses connect onto INs that form intrinsic connections within the SC (*Whyland et al., 2020*). These findings support the notion that an effective feedforward inhibition might also be at play in the retinocollicular system. But whether afferent inputs are differentially integrated by excitatory and inhibitory SC neurons, similar to the thalamocortical system, is unknown. However, our previous study in the retinocollicular system lacks the cell-type specificity, and it remains an open question whether the strong drive is specific to inhibitory SC neurons or if it is a general property of retinocollicular connections, and thus cell-type independent. Therefore, whether there are any differences in connection strengths between RGCs that project to excitatory and inhibitory SC neurons remains elusive. In this study, we investigate how EXNs and INs in the SC integrate retinal inputs in vivo.

We aim to answer these questions by characterizing the properties of the long-range synaptic connections between RGC-SC pairs employing a set of measurements: (1) The connection efficacy is used to study how strongly excitatory and inhibitory SC neurons are driven by their RGC input. (2) The functional similarity is analyzed between connected RGC-SC pairs to examine whether comparable wiring motifs govern retinocollicular signaling in EXNs and INs. (3) Short-term dynamics of retinal activity are studied to estimate whether the retinal spiking pattern shows paired-spike enhancement on retinocollicular synapses. (4) Finally, the connection contribution is employed to investigate how tightly the SC activity is coupled to retinal spiking.

This study intends to generate a more detailed picture of cell type-specific retinocollicular innervation and to fill a gap concerning the different long-range functional connectivity motifs that exist along the visual pathways.

## Results

### Recording EXNs and INs simultaneously with RGC axons in the mouse SC in vivo

To investigate the integration of retinal input in EXNs and INs in the SC in vivo, we recorded extracellular neural activity in the visual layers of SC in VGAT-ChR2 mice using high-density Neuropixels probes (*Jun et al., 2017*). We employed a tangential insertion approach to maximize the extent of visually driven channels in the SC (*Figure 1A*; *Sibille et al., 2022b*). Notably, we recently demonstrated that this approach allows us to not only measure spiking activity from SC neurons, but also to record electrical signals from RGC axons that terminate in the SC (*Sibille et al., 2022a*). The somatic

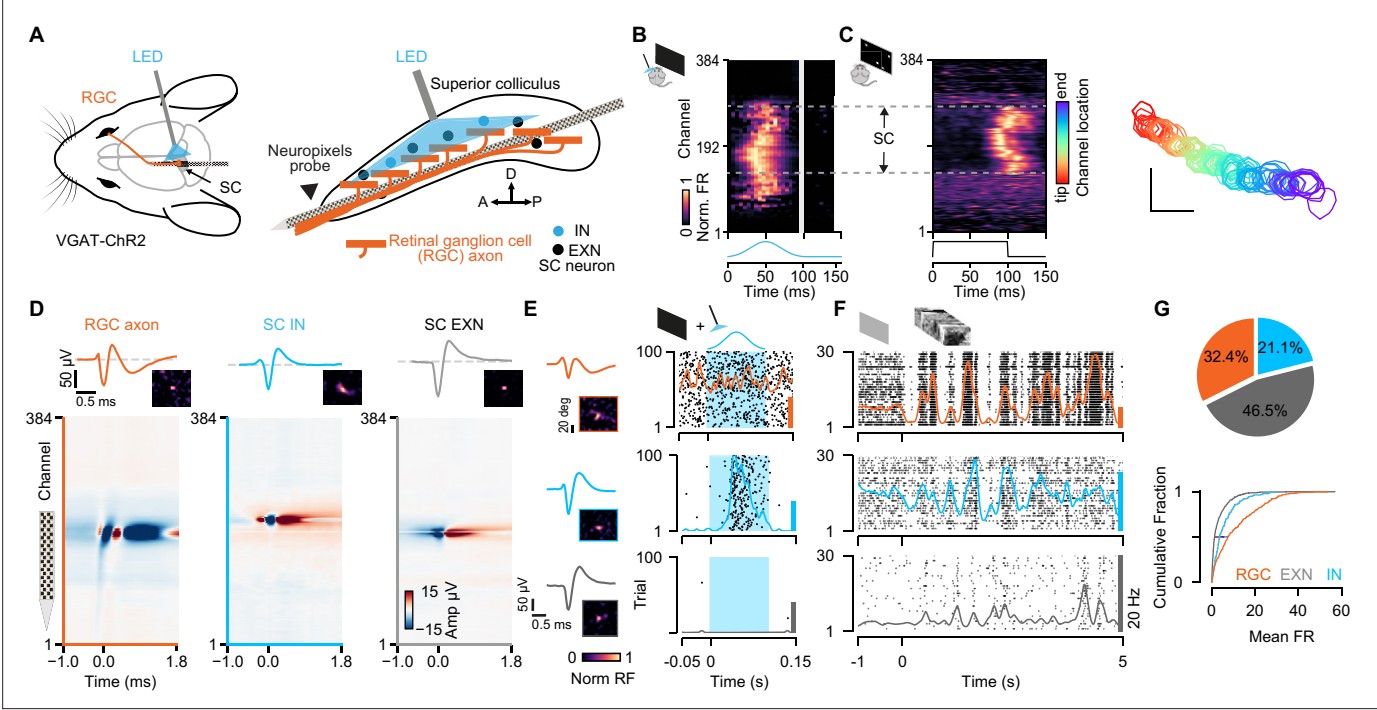

**Figure 1.** Simultaneous extracellular recordings of retinal ganglion cell (RGC) axons and superior colliculus (SC) neurons combined with optotagging identifies GABAergic neurons in VGAT-ChR2 mice. (**A**) Recording configuration for tangential electrode insertion and optotagging in the visual layers of mouse SC. The optogenetic fiber is inserted perpendicularly to the Neuropixels probe to activate GABAergic cells in VGAT-ChR2 mice. RGC afferents (orange) project onto GABAergic (blue) and non-GABAergic (black) neurons in SC. (**B**) Multi-unit response (MUA) to optogenetic stimulation along the 384 recording sites during the presentation of a black screen. The large spatial extent of the optogenetic activation is evident. Gray vertical bar = light artifacts induced by the LED stimulation. (**C**) Visually evoked MUA during the presentation of a sparse noise stimulus along the recording sites. Receptive field contours of recording sites with high signal-to-noise ratio. The color code reflects the location within the SC. Scale bar represents 10°. (**D**) Spatiotemporal waveform profiles. Single channel waveforms identified at the peak channel (top) and multi-channel waveforms (bottom) for RGC axon (left, orange) and inhibitory (middle, blue) and excitatory (right, gray) SC neurons. Receptive fields (RFs) indicate visually responsive neurons. (**E**) Identification of SC cell types via optotagging. Raster plots and peri-stimulus time histograms (PSTHs) for single-neuron responses to blue light pulses (100 ms) presented under baseline conditions. Excitatory SC neurons (EXNs, bottom) and RGCs (top) do not respond to the LED pulse, while GABAergic SC neurons respond to the light pulse with an increase in spiking response (middle). Optogenetic stimulation period is highlighted in blue. The colored scale bars on the right represent 20 Hz firing rate. (**F**) PSTHs and raster plots for different cell types shown in (E) in response to a natural movie stimulus (10 s, 30 trials). Note the high firing rate in RGCs. (**G**) Top: Proportion of identified GABAergic (INs, blue), non-GABAergic (EXNs, gray) SC neurons and retinal axons (RGCs, orange) populations. Note that around one third of the captured SC neurons are GABAergic (n=326 RGCs, n=468 EXNs, n=212 INs). Bottom: Cumulative distribution of mean firing rates in response to a natural movie stimulus presented for 10 s, 30 trials. Two-sided Wilcoxon rank-sum test.

The online version of this article includes the following source data, source code, and figure supplement(s) for figure 1:

**Source code 1.** Code to plot *Figure 1D–G*.

**Source data 1.** Data for *Figure 1G*: mean firing rates.

**Figure supplement 1.** Optogenetic response and spatial distribution of excitatory and inhibitory superior colliculus (SC) neurons.

**Figure supplement 2.** Spike waveform features analysis for GABAergic and non-GABAergic neuron populations in the superior colliculus (SC).

action potentials of SC neurons can be distinguished from the axonal action potentials of RGCs based on their spatiotemporal waveforms profile (*Sibille et al., 2022a*). RGC axonal waveforms are triphasic with a larger spatial spread (*Figure 1D*, left) while somatic SC neurons show a characteristic biphasic waveform profile (*Figure 1D*, middle and right).

To study cell type-specific differences, we combined the Neuropixels recordings with an optogenetic identification strategy (optotagging) to identify GABAergic SC neurons in VGAT-ChR2-EYFP (*Zhao et al., 2011*) mice (*Figure 1A and B*; data from n=11 experiments in n=9 mice included in the analysis, if not stated otherwise). In these transgenic mice, GABAergic neurons specifically express Channelrhodopsin-2 and can be identified upon blue light stimulation. To photo-activate a large proportion of GABAergic SC neurons using short light pulses (100 ms pulses, 2 Hz, 2.7 mW), we

inserted the optic fiber perpendicularly to the Neuropixels probe (*Figure 1A*, right, see Materials and methods). Using this recording and light stimulation configuration, we could activate GABAergic SC neurons across an average of ~1.5 mm of SC tissue (light-activated range = 1.50 ± 0.41 mm, *Figure 1B*) which also allowed us to record SC neurons along continuous retinotopy (*Figure 1C*, right) in a cell type-specific manner (*Figure 1G*). We used the visually evoked activity and continuous retinotopy to demarcate the boundaries of the SC (*Figure 1C*). Moreover, for the characterization of excitatory and inhibitory classes of SC neurons, we limited our analysis to single units that were localized within the range of optogenetic stimulation (*Figure 1B, C* and *Figure 1—figure supplement 1A*, see Materials and methods). Cell types were characterized using a custom-written graphical user interface (GUI, see Materials and methods) to label units that were activated by the LED pulse as INs (*Figure 1E*, middle). These neurons were distinguished based on their reliable increase in firing rate in response to optogenetic stimulation during spontaneous (black screen) and visually evoked conditions (checkerboard stimulus, see Materials and methods). SC neurons that were not modulated by the LED pulses, but located within the light-activated range, were labeled as EXNs (*Figure 1E* bottom). Excitatory and inhibitory SC neurons were equally distributed within the boundaries of LED stimulation for 9/11 experiments (p-values = $3.07 \times 10^{-1}$, $1.15 \times 10^{-2}$, $7.55 \times 10^{-1}$, $8.34 \times 10^{-1}$, $5.01 \times 10^{-6}$, 0.79, 0.80, 0.26, 0.33, 0.08, 0.13, Wilcoxon rank-sum test). In only two experiments EXNs and INs were not equally distributed, but only two single units were located in the region without including SC IN units, that passed the quality criteria for spike sorting (*Figure 1—figure supplement 1B*, see Materials and methods). As expected, retinal axons were not activated by the LED pulse (*Figure 1E*, top), except for a few cases (4/326 RGCs) which were excluded from the connectivity analysis in this study. In total, we recorded 326 RGC axons and 680 SC neurons. Among the recorded SC neurons one third (31.2%) were GABAergic (*Figure 1G*, n=468 EXNs, n=212 INs).

Having sorted the SC neurons into the cell classes using optogenetic stimulation (EXNs and optogenetically identified INs), we compared different spike waveform features to test whether SC neurons show differences in their waveforms. Characterizing the spike waveforms is commonly used in the cortex or hippocampus to differentiate between glutamatergic and GABAergic neurons (*Lee et al., 2016*; *Moore and Wehr, 2013*; *Niell and Stryker, 2008*). However, the separation based on differences in waveform features was not an adequate measure to separate SC EXNs from SC INs (*Figure 1—figure supplement 2*). This result is in line with findings by *Essig et al., 2021*, who did not observe a cell type-specific difference between waveform features in the SC (*Essig et al., 2021*). Similar findings have been shown in the inferior colliculus (*Ono et al., 2017*), a neighboring midbrain structure, suggesting that waveform classification analyses used in the cortex are not suitable for distinguishing neurons in the midbrain.

To further characterize the response properties of the different cell classes (RGC, SC INs, SC EXNs), we measured their firing rates in response to a natural movie stimulus provided in *Froudarakis et al., 2014* (see Materials and methods). We found that RGCs show the highest firing rate (firing rate RGC: median = 7.55 spikes/s, Q1=2.39 spikes/s, Q3=17.93 spikes/s, n=326 RGCs; *Figure 1G*, bottom). On average, the mean firing rate in SC INs (firing rate IN: median = 3.63 spikes/s, Q1=1.6 spikes/s, Q3=8.02 spikes/s, n=212 INs) was 1.8 times higher compared to that of SC EXNs (firing rate EXN: median = 1.38 spikes/s, Q1=0.49 spikes/s, Q3=4.19 spikes/s, n=468 EXNs; SC EXN vs SC IN p=$3.1567 \times 10^{-15}$, SC EXN vs RGC p=$4.8362 \times 10^{-40}$, SC IN vs RGC p=$2.7388 \times 10^{-8}$, two-sided Wilcoxon rank-sum test). Our results are in line with previous studies by *Usrey et al., 1998*, showing that RGCs show high mean firing rates (*Kara and Reid, 2003*; *Usrey et al., 1998*). Moreover, from other midbrain areas it is also known that GABAergic neurons show higher firing rates compared to glutamatergic cells (*Ono et al., 2017*).

In summary, by combining Neuropixels recordings to measure the activity of RGC axons and SC neurons with optogenetic identification of GABAergic SC neurons, it becomes possible to measure monosynaptic retinocollicular connectivity in vivo and retinal innervation strength in a cell type-specific manner.

## Retinal input integration in excitatory and inhibitory SC neurons

To study the monosynaptic retinocollicular connectivity in vivo and retinal innervation strength, we identified putative monosynaptically connected RGC-SC pairs employing established cross-correlogram (CCG) analysis methods (*Alonso et al., 2001*; *Bereshpolova et al., 2020*; *Sibille et al., 2022a*). A

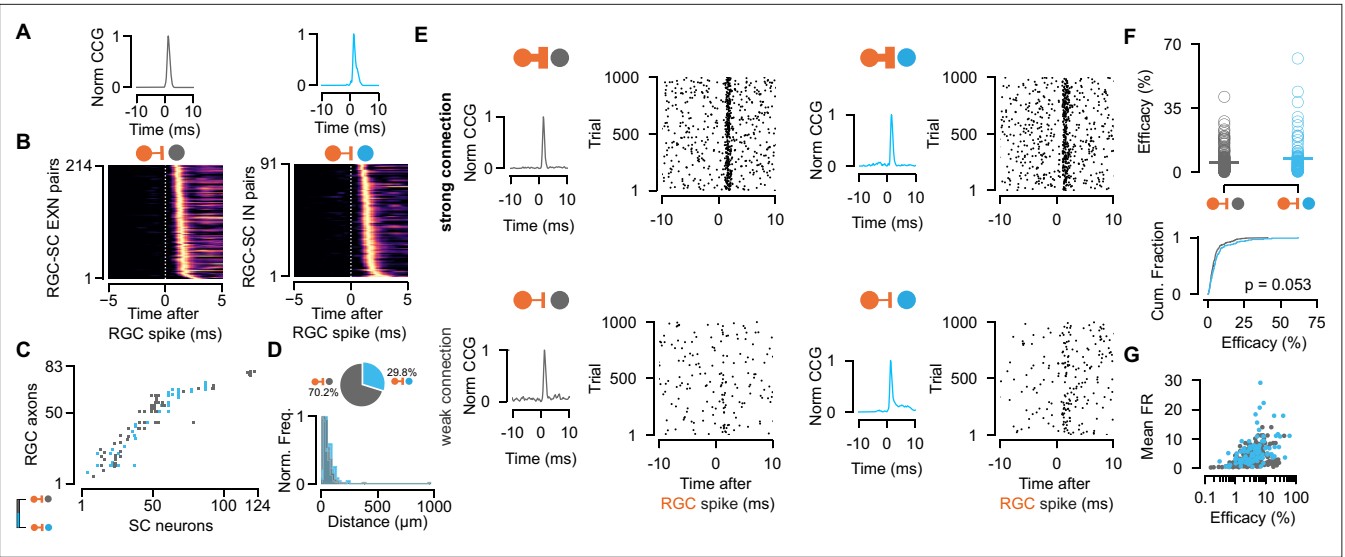

**Figure 2.** Retinal innervation is similarly strong to excitatory and inhibitory superior colliculus (SC) neurons. (**A**) Monosynaptically connected retinal ganglion cell (RGC)-SC excitatory neuron (EXN) (gray) and RGC-SC inhibitory neuron (IN) (blue) pairs are identified via cross-correlogram (CCG) analysis. (**B**) CCGs of connected RGC-SC EXN and RGC-SC IN pairs sorted by their peak latency (n=214 RGC-SC EXN, n=91 RGC-SC IN, n=11 recordings). (**C**) Connectivity matrix from a single recording. Gray marks indicate connections onto excitatory SC neurons, blue marks indicate connections onto inhibitory SC neurons. RGC axons and SC neurons are sorted by their peak channel along the electrode. (**D**) Distribution of peak channel distances between RGC axons and connected SC neurons (p=0.0328, two-sided Wilcoxon rank-sum test). Inset shows pie chart of identified RGC-SC IN and RGC-SC EXN pairs. (**E**) Elicited SC spiking in response to firing of a presynaptically connected retinal ganglion cell (RGC). Raster plot shows SC firing to 1000 randomly selected RGC spikes. Both SC cell types show robust activation upon RGC spiking (top) but also weaker connections can be found (bottom). (**F**) Scatter plot and cumulative distribution of synaptic efficacy as a measure for connection strength for RGC-SC EXN and RGC-SC IN connected pairs (p=0.053, n=214 RGC-SC EXN, n=91 RGC-SC IN). (**G**) Efficacy as a function of mean firing rate (FR) during the entire recording session (RGC-SC EXN $r$=0.42685; p<0.0005; RGC-SC IN: $r$=0.43543, p=0.00002). Two-sided Wilcoxon rank-sum test.

The online version of this article includes the following source data for figure 2:

**Source code 1.** Code to plot *Figure 2D, F, G*.

**Source data 1.** Data for *Figure 2D* describing the distance between retinal ganglion cell (RGC) axon and superior colliculus (SC) neuron on the electrode probe; and *Figure 2F, G* efficacy values.

significant transient and a short-latency peak in the spike train CCG is a hallmark of monosynaptic connectivity (*Figure 2A*) and identifies connected RGC-SC pairs. We measured up to 98 monosynaptic connections in individual recordings, depending on the number of measured RGC axons (*Figure 2C*). In total, we identified 305 connected RGC-SC pairs, of which 29.8% were retinal axons that connected onto GABAergic SC neurons (n=214 connected RGC-SC EXN pairs, n=91 connected RGC-SC IN pairs, *Figure 2B, D*). This high yield in monosynaptically connected pairs was due to the close proximity of the simultaneously recorded RGC axons and SC neurons on the high-density electrode (*Figure 2D*, bottom) (*Sibille et al., 2022a*). For most of the connected RGC-SC pairs the peak channels on the electrode were on average 50–60 μm apart on the probe and hence within SC (distance RGC-EXN: median = 40.0 μm, first quartile = 25.61 μm, third quartile = 62.1 μm; distance RGC-IN: median = 51.22 μm, first quartile = 32.0 μm, third quartile = 83.08 μm). We found a significant difference in peak channel distance between RGC-SC EXN and RGC- SC IN pairs with longer distances between RGC axons and inhibitory SC neurons (p=0.03287, two-sided Wilcoxon rank-sum test), however, the effect size was small (Cohen's d=0.0915). The maximum distance for retinal axons and inhibitory SC neurons was 240 μm, while the peak channels of one RGC-SC EXN pair were 960 μm apart. The latter pair might include an SC wide-field (WF) neuron as these cells are excitatory and their dendritic arbor widths have been reported to spread up to 900 μm (*Gale and Murphy, 2014*).

To assess the connection strength between RGCs and excitatory and inhibitory SC neurons, we calculated the connection efficacy (*Usrey et al., 1999*). The efficacy measures the connection strength as the probability of an RGC spike triggering an action potential in the postsynaptic SC neuron (see Materials and methods). We found highly diverse connectivity patterns for RGC-SC EXC and RGC-SC

IN pairs. RGC activity evoked robust firing in both SC cell types (*Figure 2E*, top), however, pairs with weak connection strength were also observed as recently described in *Sibille et al., 2022a*; *Figure 2E*, bottom. Across the population, both SC cell types received similarly strong input from the retina (efficacy RGC-SC IN: median = 4.54 %, Q1=2.2 %, Q3=7.96 %, maximum = 62.05 %, n=91 connected pairs; efficacy RGC-SC EXN: median = 3.5 %, Q1=1.96 %, Q3=5.89 %, maximum = 41.12 %, n=214 connected pairs; *Figure 2F*). While we observed a tendency for slightly stronger RGC-SC IN connections, this difference was not statistically significant (p=0.053, two-sided Wilcoxon rank-sum test). To assess whether the connectivity strength is related to the firing of the postsynaptic cell, we examined the relationship between the efficacy and the overall firing rate throughout the entire recording session. Surprisingly, our results demonstrate a positive correlation between the efficacy and the mean firing rate on the population level (*Figure 2G*, RGC-SC EXN $r$=0.42685; p<0.0005; RGC-SC IN: $r$=0.43543, p=0.00002, two-sided Wilcoxon rank-sum test).

Here, we set out to explore how RGC spikes drive the activity in postsynaptic excitatory and inhibitory SC neurons. We did not find a cell type-specific difference in connection strength and therefore conclude that inhibitory and excitatory SC neurons are innervated comparably strong and are both reliably driven by their retinal inputs.

## Functional similarity of retinocollicular connections

In sensory cortices, INs are less selective and receive a diverse and nonspecific set of thalamic inputs compared to EXNs which receive more specific thalamic inputs (*Alonso and Swadlow, 2005*; *Bruno and Simons, 2002*). However, the selectivity and functional similarity between RGCs and their excitatory and inhibitory target populations in SC remains largely unknown (*Shi et al., 2017*). Despite both

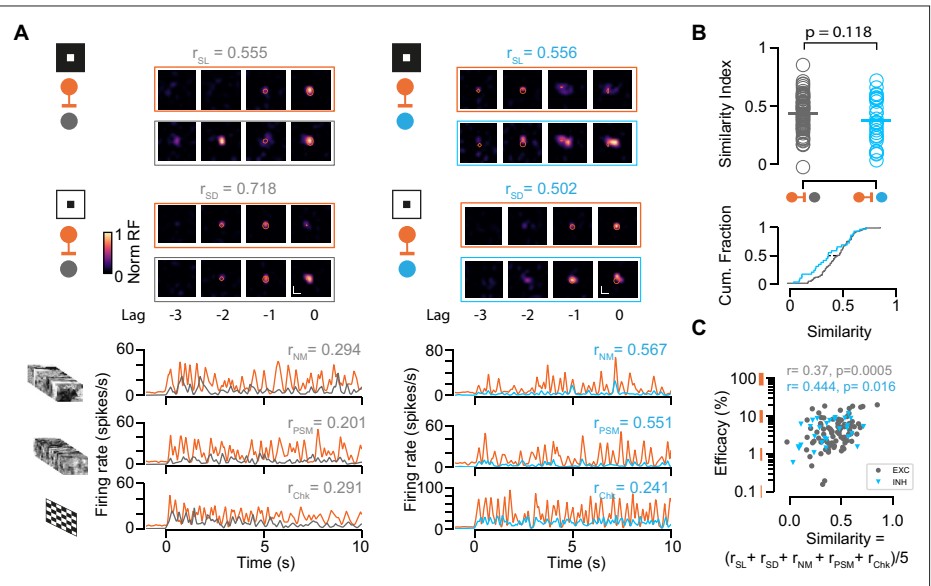

**Figure 3.** Characterization of functional similarity between retinocollicular connected pairs. (**A**) Top: Spatiotemporal receptive fields (STRF) evoked by dark (SD) and light (SL) sparse noise stimuli for retinal ganglion cell (RGC)-superior colliculus (SC) excitatory neuron (EXN) and RGC-SC inhibitory neuron (IN) connected pairs. Bottom: Visually evoked activity in response to a natural movie (NM), phase scrambled movie (PSM), and dense noise (Chk) stimulus. The functional similarity of the RGC axon and the postsynaptic SC neuron is characterized by the correlation coefficient during the different stimuli conditions ($r_{SD}$, $r_{SL}$, $r_{NM}$, $r_{PSM}$, $r_{Chk}$). (**B**) The overall functional similarity between the presynaptic RGC and the postsynaptic SC neurons is reflected in the similarity index calculated from the averaged correlation coefficients ($r_{SL}$ + $r_{SD}$ + $r_{NM}$ + $r_{PSM}$ + $r_{Chk}$)/5 (p=0.118, two-sided Wilcoxon rank-sum test, n=85 RGC-SC EXN pairs, n=29 RGC-SC IN pairs). Scatter plot (top) and cumulative distribution (bottom) of similarity index. (**C**) Relationship between similarity index and connection efficacy (Pearson correlation coefficient; n=85 RGC-SC EXN pairs, n=29 RGC-SC IN pairs).

The online version of this article includes the following source data for figure 3:

**Source code 1.** Code to plot *Figure 3B, C*.

**Source data 1.** Data for *Figure 3B*: similarity index and *Figure 3C*: efficacy values.

SC populations showing a similar innervation strength from retinal axons, INs could sample from a more functionally diverse population of RGCs. To test this, we characterized the functional similarity between the RGCs and their connected postsynaptic SC neurons. To this end, we calculated the correlation coefficients between the trial-averaged visually evoked activity of connected RGC-SC pairs in response to light and dark sparse noise stimuli ($r_{SL}$ and $r_{SD}$) as well as the natural movie stimulus ($r_{NM}$), a phase scrambled version ($r_{PSM}$), and a checkerboard stimulus ($r_{Chk}$) (*Figure 3A/B*, RGC-SC with high signal-to-noise in their receptive field [RF], see Materials and methods). To quantify the overall functional similarity by a single similarity value, we averaged the five correlation measurements (similarity = ($r_{SL}$ + $r_{SD}$ + $r_{NM}$ + $r_{PSM}$ + $r_{Chk}$)/5). A similarity value of 1 corresponds to perfectly correlated visually driven responses between the RGC and the postsynaptic SC neuron, while a value of 0 signifies uncorrelated responses. In our data, the RGC-SC pairs span a wide range of functional similarity, however, we found no significant difference between RGC-SC EXN and RGC-SC IN pairs (similarity for RGC-SC EXN = 0.44 ± 0.15 and RGC-SC IN = 0.37 ± 0.19, n=85 RGC-SC EXN pairs, n=29 RGC-SC IN pairs, p=0.245, two-sided Wilcoxon rank-sum test, *Figure 3B*). In a next step, we correlated the connection efficacy measure to the similarity index. The synaptic efficacy was positively correlated with the functional similarity of the connected pairs, with a stronger correlation for RGC-SC IN pairs (RGC-SC EXN r=0.37, p=0.0005; RGC-SC IN r=0.44, p=0.02, *Figure 3C*). Overall, we observed that functionally similar pairs are more strongly connected. However, we also observed cases of relatively strong connected RGC-SC pairs (~10%) with low similarity (*Figure 3C*), suggesting that some SC neurons receive convergent input from a functionally more diverse pool of RGC afferents (*Sibille et al., 2022a*). Our findings delineate that both RGC-SC EXN and RGC-SC IN connected pairs are organized, while

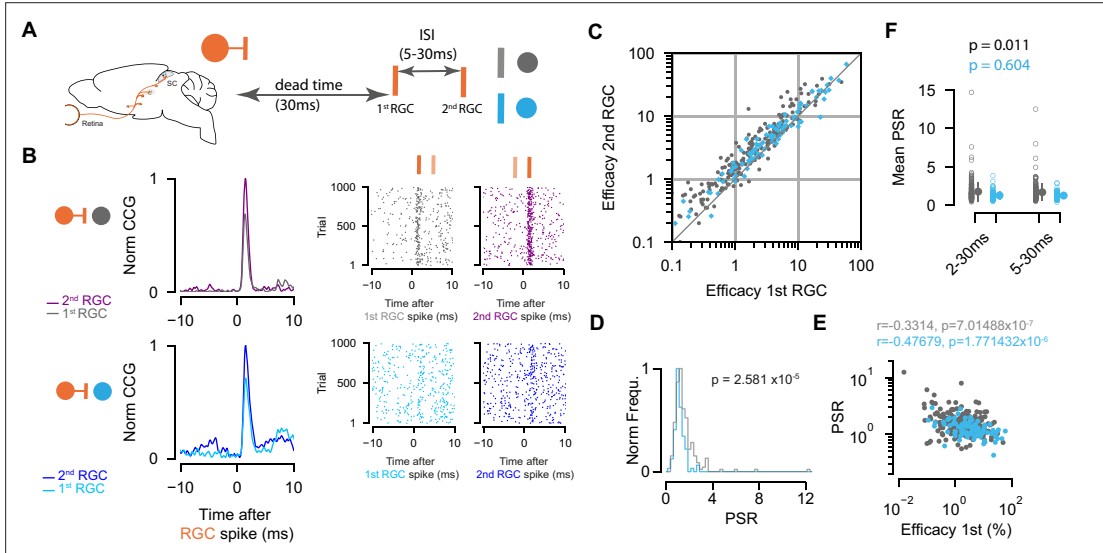

**Figure 4.** Paired-spike dynamics: Second retinal spikes are more efficient in driving superior colliculus (SC) response. (**A**) Schematic illustrating the temporal dynamics between two successive retinal ganglion cell (RGC) spikes. Pairs of RGC spikes with a minimum inter-spike interval (ISI) of 5 ms and maximum ISI of 30 ms were included if there were no preceding spikes before the first RGC for a dead time of at least 30 ms. (**B**) Cross-correlograms (CCGs) of example RGC-SC excitatory neuron (EXN) and RGC-SC inhibitory neuron (IN) pairs calculated from spike trains selected for first and second RGCs and the corresponding raster plots to 1000 trials triggered on RGC spikes. (**C**) Scatter plot of efficacies for first versus second retinal spikes. The majority of connected pairs (185/214 RGC-SC EXN pairs; 73/91 RGC-SC IN pairs) showed paired-spike enhancement in response to second RGC spikes. (**D**) Distribution of paired-spike ratio (PSR) for RGC-SC EXN and RGC-SC IN connected pairs. The paired-spike enhancement is stronger in SC-EXNs (n=214 RGC-SC EXN; n=91 RGC-SC IN, p=2.581 × 10⁻⁵, two-sided Wilcoxon rank-sum test). (**E**) Correlation of first RGC efficacy with the PSR for connected pairs (r=0.3314 for n=214 RGC-SC EXN pairs and r=0.4767 for n=91 RGC-SC IN pairs; logarithmic values; Pearson correlation coefficient test). (**F**) Change in mean PSR depending on ISI for different groups of ISI (2–3 ms and 5–30 ms). The 2–30 ms group ensures to include high-firing RGCs (RGC-SC EXN pairs 2–30 vs. 5–30 ms p=0.011; RGC-SC IN pairs 2–30 vs. 5–30 ms p=0.604; Wilcoxon signed-rank test).

The online version of this article includes the following source data for figure 4:

**Source code 1.** Code to plot *Figure 4C–F*.

**Source data 1.** Data for *Figure 4C*: efficacy values in response to first and second retinal ganglion cell (RGC) spikes, *Figure 4D, E*: paired-spike ratio (PSR) values, and *Figure 4F*: PSR values for different groups.

functionally similar properties between the pre- and postsynaptic neuron are reflected in a stronger connection.

## Paired-spike interactions of the retinocollicular connections

We showed that retinal spikes are efficient in driving SC activity for both EXNs and INs. As the timing of presynaptic activity has been assessed as a parameter that modulates postsynaptic firing (*Zucker, 1989*) and connection strength (*Kara and Reid, 2003*; *Usrey et al., 1998*), we wanted to know whether the temporal pattern of retinal activity has different effects on the synaptic efficacy of excitatory and inhibitory SC neurons. To that end, we studied the short-term dynamics (facilitation or depression) of the measured connections and performed a paired-spike analysis approach as described by *Usrey et al., 1998*. A pair of retinal spikes (first, second RGC) was defined by its temporal firing patterns: inter-spike interval (ISI) and dead time. We selected RGC spikes for which a second RGC spike followed the first RGC spike by a specific ISI (minimum ISI = 5 ms, maximum ISI = 30 ms). The dead time ensures that the first RGC was preceded by a minimum period without activity (30 ms) (*Figure 4A*).

We found that overall second retinal spikes were more efficient in driving SC activity, and thus facilitating. This was true for both RGC-SC EXN (efficacy first RGC: median = 1.8 %, Q1=0.77 %, Q3=4.47%; efficacy second RGC: median = 2.66 %, Q1=1.16 %, Q3=6.07 %, p first vs second=$3.13 \times 10^{-25}$, n=214 pairs) and RGC-SC IN connected pairs (efficacy first RGC: median = 3.66 %, Q1=1.6, Q3=8.61; efficacy second RGC: median = 4.33, Q1=2.03, Q3=8.25, p first vs second=$1.68 \times 10^{-6}$, n=91 pairs, Wilcoxon signed-rank test) (*Figure 4B and C*). This effect is known as paired-spike enhancement and has been reported earlier in retinogeniculate (*Mastronarde, 1987*; *Usrey et al., 1998*) and geniculocortical (*Usrey et al., 2000*) connections. The majority of retinocollicular connected pairs showed paired-spike enhancement where second retinal spikes were more effective in driving SC response (86.45% [185/214] RGC-SC EXN pairs, 80.22% [73/91] RGC-SC IN pairs). The effect was more pronounced in RGC-SC EXN pairs where the second retinal spike was 1.7 times as effective in driving responses in the postsynaptic SC neuron, while for RGC-SC IN pairs, the second retinal spike was only 1.3 times more efficacious in driving SC spiking. To quantify this observation, we calculated the paired-spike ratio (PSR = efficacy second/efficacy first) which confirmed a higher facilitation effect for RGCs connecting onto EXNs (PSR RGC-SC EXN = 1.711 ± 1.14; RGC-SC IN: 1.281±0.414, p=$2.58 \times 10^{-7}$, Wilcoxon rank-sum, n=305 pairs, *Figure 4D*).

Previous work in the thalamocortical system has shown that weaker synaptic connections can be more strongly modulated compared to stronger synaptic connections (*Ferrarese et al., 2018*). To test the notion that also in the retinocollicular system, connected pairs with lower efficacy in response to first RGCs show greater enhancement in response to the second RGCs, we correlated the efficacy of the first RGC to the PSR (*Figure 4E*). We observed a negative correlation between the logarithm of the first RGC efficacy and the logarithm of PSR for both RGC-SC EXN (r=0.3314, p=$7.0148 \times 10^{-7}$, n=214 pairs) and RGC-SC IN pairs (r=0.4767, p=$1.7714 \times 10^{-6}$, n=91 pairs). However, we found pairs that show low efficacy in response to the first RGC and low PSR. Taken together, we observed paired-spike enhancement in RGCs connecting onto both inhibitory and excitatory SC neurons, with a stronger facilitation effect for retinal afferents connecting onto excitatory, in comparison to inhibitory, SC target neurons. To estimate whether the ISI has an effect on the PSR, we added another group of 2–30 ms ISI to include RGCs that fire at high frequencies (PSR RGC-SC EXN = 1.783 ± 1.23; RGC-SC IN: 1.295±0.528, p=$7.03 \times 10^{-6}$, Wilcoxon rank-sum, n=305 pairs, *Figure 4F*). However, we did not observe a clear difference between the 2–30 ms and 5–30 ms groups for inhibitory connections (SC IN: p=0.604, Wilcoxon signed-rank test). In contrast, the two different ISI groups of EXNs show a statistically significant difference (p=0.011, Wilcoxon signed-rank test), however, the effect size is small (Cohen's d RGC-SC EXN = 0.063, RGC-SC IN = 0.030).

## Inhibitory SC neurons are less coupled to the retina compared to excitatory SC neurons

Our data so far reveals that excitatory and inhibitory SC neurons receive similarly strong drive from their retinal afferents. However, we observed a higher mean firing rate in SC INs compared to SC EXNs, on average (*Figure 1G*), which is intriguing given the similarity in the RGC drive on both cell types. A possible explanation could be that the activity of SC INs is less coupled to the RGC inputs

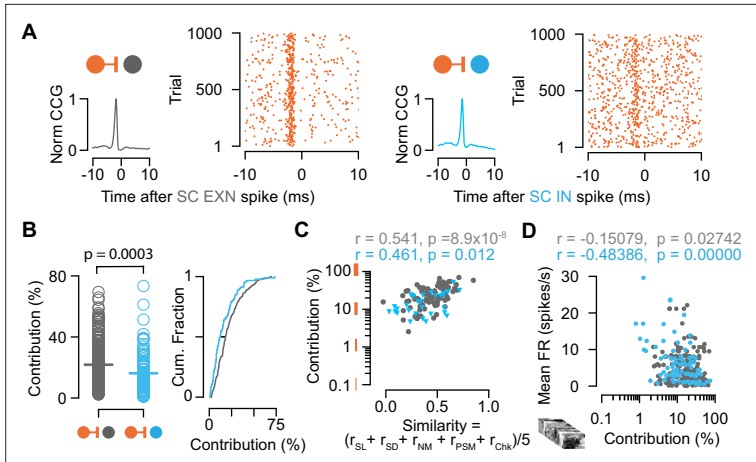

**Figure 5.** Connection contribution is higher in connected retinal ganglion cell (RGC)-superior colliculus (SC) excitatory neuron (EXN) pairs. (**A**) Example cross-correlograms (CCGs) of monosynaptically connected RGC-SC EXN (left) and RGC-SC inhibitory neuron (IN) (right) pairs. Connected pairs were identified by their peaks in the CCGs (left). Raster plots of RGC spiking activity triggered on 1000 SC spikes (right). (**B**) Scatter plot and cumulative distribution of contribution values for RGC-SC EXN and RGC-SC IN pairs (n=214 RGC-SC EXN pairs, n=91 RGC-SC IN pairs, p=0.0003, two-sided Wilcoxon rank-sum test). (**C**) Relationship between functional similarity index and connection contribution (Pearson correlation coefficient test; n=85 RGC-SC EXN pairs, n=29 RGC-SC IN pairs). (**D**) Correlation of mean firing rate (to natural movie stimulus) and contribution for RGC-SC EXN and RGC-SC IN pairs (Pearson correlation coefficient, two-sided Wilcoxon rank-sum test, n=214 RGC-EXN and n=91 RGC-IN connected pairs; n=11 penetrations from nine mice).

The online version of this article includes the following source data, source code, and figure supplement(s) for figure 5:

**Source code 1.** Code to plot *Figure 5B–D*.

**Source data 1.** Data for *Figure 5B, C, and D*: contribution values, mean firing rates, and similarity index.

**Figure supplement 1.** Contribution as a function of firing rate during different stimulus conditions.

as compared to that of SC EXNs, that is, SC INs spikes are less often driven by their RGC inputs. To examine how strongly the activity of the two populations of SC neurons is coupled to the activity of individual RGC inputs, we estimated the synaptic coupling strength from the connection contribution (***Levick et al., 1972***; ***Sibille et al., 2022a***; ***Usrey et al., 1999***) of the measured retinocollicular pairs. The connection contribution reflects the fraction of postsynaptic SC spikes that are preceded by a presynaptic retinal spike within a short time window (***Figure 5A***). High contribution values indicate that SC neurons are primarily driven by individual RGC afferent inputs, while low contribution values point toward an integration of multiple inputs.

Interestingly, we found that the contribution of individual retinal spikes on SC activity was higher for excitatory compared to inhibitory SC neurons (***Figure 5B***) (contribution RGC-SC EXN: median = 18.27% Q1=10.79%, Q3=29.69%, n=214; contribution RGC-SC IN: median = 12.2%, Q1=6.61%, Q3=23.02%, n=91 connected pairs, p=0.0003, two-sided Wilcoxon rank-sum test). This opens up the possibility that SC EXNs might be more specifically linked to the retinal input compared to SC INs. To assess the relationship between the coupling strength and the functional similarity, we correlated both measures and found a positive correlation, suggesting that stronger coupled pairs show higher similarity (RGC-SC EXN $r$=0.541, p=8.9 × $10^{-8}$, n=85 pairs; RGC-SC IN $r$=0.461, p=0.012, n=29 pairs, Pearson correlation coefficient test, ***Figure 5C***). Moreover, we observed a diversity of firing rates in both SC EXNs and SC INs (***Figure 1G***), which suggests that low contribution values are related to the mean firing rates of the SC neurons. And indeed, when we correlated the contribution to the mean firing rate, we found a negative correlation for both RGC-SC EXN and RGC-SC IN pairs (RGC-SC EXN $r$=–0.15079, p=0.02742; RGC-SC IN $r$=–0.48386, p<0.00001, Pearson correlation coefficient test n=214 RGC-EXN; n=91 RGC-IN connected pairs) (***Figure 5D*** and ***Figure 5—figure supplement 1***).

Taken together, the cell type-specific difference in retinal coupling strength suggests that retinal input contributes a substantial part to the spiking activity of excitatory SC neurons, while inhibitory

SC neurons seem to also integrate inputs from other sources. This finding denotes that excitatory SC neurons are more tightly linked to the retinal input compared to inhibitory SC neurons.

## Discussion

In this study, we elucidate how inhibitory and excitatory SC neurons integrate visual information from the retina in vivo. By combining the tangential Neuropixels recording approach with optotagging techniques, we were able to measure RGC-SC connected pairs in a cell type-specific manner and study their functional organization. Our findings allow the following conclusions: (1) The retinal drive is similarly strong for RGCs connecting onto GABAergic and non-GABAergic SC neurons. (2) The connection strength between the RGCs and both SC cell classes is positively correlated with their functional similarity, while still allowing for the presence of strong connections with low functional similarity. (3) Paired-spike enhancement in RGC-SC connections was a signature for both excitatory and inhibitory SC neurons. (4) Finally, the contribution of a retinal spike to overall SC firing is higher in excitatory SC neurons compared to inhibitory SC neurons.

We studied retinocollicular connectivity strength and aimed to answer the question whether excitatory and inhibitory SC populations use similar or different coding schemes to integrate visual information compared to the thalamocortical visual system. Connectivity patterns and circuit motifs have been extensively studied in the thalamocortical system, yet our understanding of the encoding mechanisms that govern retinocollicular signaling remains rudimentary albeit recent studies suggest that feedforward inhibition might be at play (*Villalobos et al., 2018*; *Whyland et al., 2020*). While cortical INs are more strongly activated by thalamic axons compared to EXNs (*Bruno and Simons, 2002*; *Cruikshank et al., 2007*; *Ji et al., 2016*), our results now show that in the SC, the afferent retinal drive is comparably strong on EXNs and INs. This finding is intriguing and suggests that the two major visual brain structures in the vertebrate brain, the visual cortex and the SC, integrate afferent sensory information differentially.

Several parameters for example the degree of convergence (*Kara and Reid, 2003*), response properties (*Alonso et al., 2001*), synaptic mechanisms (*Cruikshank et al., 2007*), or spiking patterns (*Usrey et al., 1998*) can have meaningful effects on modulating the input strength on the target cell. SC neurons receive input from a small number (4–6) of RGCs (*Chandrasekaran et al., 2007*; *Furman et al., 2013*). In contrast, cortical neurons receive numerous (approximately one order of magnitude more) thalamic inputs that are usually weak (*Bruno and Sakmann, 2006*; *Jin et al., 2011*; *Lien and Scanziani, 2018*). Overall, we observed strong retinal drive onto both SC populations indicated by high efficacy values (*Figure 2*), and thus individual inputs may be sufficiently strong to drive the SC neuron. This is in contrast to thalamocortical inputs, where individually efficacy values are low, but the thalamic drive is boosted by synchronously active thalamic inputs (*Alonso et al., 1996*; *Bruno and Sakmann, 2006*). Our recording method allows for the simultaneous recording of multiple RGC afferents from SC neurons making it possible to evaluate how the synchrony among RGC afferents modulates the firing of postsynaptic SC neurons in future studies.

The difference in connection strength between cortical EXNs and INs could be partly attributed to their differences in response properties. Cortical INs display higher sensitivity, receive more convergent thalamic input, have higher firing rates, and sample more randomly while EXNs receive highly functional specific thalamic input (*Clay Reid and Alonso, 1995*; *Kyriazi et al., 1994*; *Simons and Carvell, 1989*). While we observed higher firing rates in inhibitory SC neurons (*Figure 1G*), the efficacy of the RGC-SC connections was comparable between both types of neurons (*Figure 2F*), suggesting that the sensitivity of both neuron types is not markedly disparate. In general, visual response properties between EXNs and INs in the SC may not differ to such a great extent (*Inayat et al., 2015*; *Kasai and Isa, 2016*; *Shi et al., 2017*, but see *Barchini et al., 2018*; *Li and Meister, 2022*), and experiments in the inferior colliculus show that GABAergic and glutamatergic cells share similar response properties (*Ono et al., 2017*). These observations imply that in the SC and other parts of the midbrain, the cell type-specific difference in terms of selectivity might not be as strongly separated as in the cortex. Future studies are required to deeper investigate the differences in response properties of the subclasses of SC neurons and whether these properties are correlated to the RGC-SC connection strength. The positive correlation between efficacy and overall mean firing rate (*Figure 2G*) is intriguing. It suggests that highly excited SC neurons may transmit RGC inputs more efficaciously. Interestingly, *Jouhanneau et al., 2018*, reported a negative correlation between the distance of the

excitatory postsynaptic potential peak to the action potential threshold and the efficacy (synaptic gain) in cortical PV neurons (*Jouhanneau et al., 2018*). Unfortunately, due to the inherent limitations of our extracellular recording approach, we are unable to investigate this aspect in our dataset. These findings underscore the complexity of synaptic transmission and wiring specificity and the multiple factors by which it can be influenced.

In our results, we observed an overall trend that functionally similar RGC-SC pairs show stronger connections which was true for both excitatory and inhibitory pairs (*Figure 3*). However, we also found pairs with considerably strong retinocollicular connectivity (efficacy values >10%) albeit the functional similarity between these connected cells was low (*Figure 3C*). This observation suggests that SC neurons may either show high projection specificity by integrating inputs from a functional homogeneous population of RGCs (*Li and Meister, 2022*; *Shi et al., 2017*; *Sibille et al., 2022a*) or receive diverse inputs from multiple functional RGC types (*Li and Meister, 2022*; *Sibille et al., 2022a*). Moreover, a high degree of projection specificity was recently shown to be maintained even disynaptically toward brain regions downstream of the SC (*Reinhard et al., 2019*), suggesting that at least one important role of the RGC-SC projection is to preserve the representation of the functionally diverse retinal channels (*Baden et al., 2016*). In our study, we investigated the functional similarity between connected RGC-SC pairs in response to sparse and dense noise stimuli as well as natural movie stimuli. And while we observed a diversity among the similarities between RGCs and connected targets in SC, supporting the idea that SC neurons may follow diverse wiring motifs (*Sibille et al., 2022a*), we observed no clear difference between inhibitory and excitatory SC neurons (*Figure 3B*). To further address the influence of functional properties on wiring mechanisms, it would be interesting to target the pool of distinct RGC types and to study how strongly they recruit SC subpopulations and modulate postsynaptic responses of their targets.

Temporal dynamics of presynaptic activity patterns are known to play a role in modulating synaptic strength (*Usrey et al., 1998*; *Zucker, 1989*). To study how the temporal spiking pattern of the RGC activity shapes RGC-SC connectivity, we analyzed the paired-spike dynamics. Across the population, we found a facilitating effect where responses to second retinal spikes were more efficient in driving SC activity for both SC neuron populations. However, the paired-spike enhancement was more pronounced for EXNs than INs (*Figure 4*). Moreover, our observation highlights differences in the encoding strategies used by different regions of visual processing. While retinogeniculate connections show paired-spike enhancement (*Usrey et al., 1998*), thalamocortical and corticotectal synapses express mostly depressing dynamics (*Bereshpolova et al., 2006*; *Swadlow et al., 2002*; *Swadlow and Gusev, 2001*, but see *Usrey et al., 2000*). Our data now reveals that retinocollicular connections facilitate, suggesting that the temporal integration of RGC inputs by SC neurons may be similar to the visual thalamus (dorsal lateral geniculate nucleus [dLGN]). The dLGN serves as another main target of RGC axons, however, in the mouse, the dLGN mainly consists of relay neurons, and only approximately 6% of INs (*Butler, 2008*; *Evangelio et al., 2018*). This disparity in interneuron density makes it challenging to directly compare the SC and dLGN circuits. Despite this difference, there exists a common synaptic motif between the dLGN and the SC. RGC axons innervating the dLGN form triadic arrangements with EXNs and INs (*Sherman, 2004*; *Wilson et al., 1984*) and this arrangement can also be found in the SC (*Whyland et al., 2020*). It has been shown that GAD67 is a suitable marker to target the INs involved in the triadic motif in the SC (*Whyland et al., 2020*) and therefore, it would be interesting to study retinal connectivity in the context of these triads in the SC using GAD67 mice.

In our study, we used CCGs that were calculated from spike trains throughout the full recording period, to be applied on different connectivity strength measures. However, paired-spike dynamics are known to be modulated by the local temporal structure of inputs, for example the visual stimulus presented (*Usrey et al., 1998*). In addition, facilitating and depressing effects can change with the target population (*Bereshpolova et al., 2006*). Further studies are required to investigate whether the facilitating effect observed in our study changes upon targeting defined collicular subpopulations that may have differences in firing rates or during different stimulus conditions (e.g. spontaneous and stimulus evoked). In this study, retinocollicular connectivity was measured in anesthetized conditions. For future studies it would be interesting to examine connectivity dynamics and innervation strength in awake conditions as it has been shown that retinocollicular synapses are modulated by the animals' level of arousal (*Schröder et al., 2020*). Likewise, more than 30 different types of RGCs have been characterized (*Baden et al., 2016*; *Sanes and Masland, 2015*) and understanding how specific

RGC types relay information onto the different SC neuron types would be important to address in more detail in future studies. Moreover, previous studies have shown a relationship between the response properties of SC neurons and their location along the dorsal/ventral axis (*Liu et al., 2023*) and differences in the density and size of retinocollicular terminals along this axis (*Carter et al., 1991*). Exploring this aspect further utilizing a vertical recording approach could provide valuable insights in the context of functional organization.

Despite our observation that retinal drive is similarly strong onto both SC cell types, the contribution of retinal activity to individual SC neurons firing was higher in excitatory SC neurons. This opens up the possibility that excitatory SC neurons may be more specifically linked to retinal inputs, while inhibitory SC cells might receive excitation from other sources aside from the retina. This finding raises the question on what other sources could drive spiking in inhibitory SC neurons?

1. RGCs provide the main input to the superficial SC (*Ellis et al., 2016*) with limited convergence (*Sibille et al., 2022a*). Nonetheless, the convergence may be higher for RGCs connecting onto inhibitory SC neurons which could explain the lower contribution and higher mean firing rates we found in INs. However, it has to be considered that our in vivo recording technique is limited in capturing the total pool of converging RGC inputs onto a neuron (*Sibille et al., 2022a*). Moreover, it has been shown that different types of RGCs exhibit variations in their axonal diameter, and this diameter is associated with how these cells relay information (*Perge et al., 2009*). As a result, our recording approach may have a tendency to capture signals from RGCs with high firing rates that are located in close proximity to the Neuropixels probe. Viral retrograde transsynaptic tracing methods (*Muellner and Roska, 2023*; *Rompani et al., 2017*) could be employed to study the full presynaptic RGC pool of single SC neurons.
2. In addition to retinal inputs, the SC receives extensive projections from the visual cortex (V1) that could act as a potential candidate to excite INs. And while indeed GABAergic horizontal cells have been shown to receive V1 input (*Zingg et al., 2017*), the majority (93%) of cortico-tectal terminals target the dendrites of non-GABAergic neurons in the SC (*Masterson et al., 2019*). Therefore, more work is needed and studying V1 excitation to defined inhibitory SC subpopulations will be an important topic to be investigated by the use of tracing studies and will shed light on this complexity.
3. Intracollicular connections can provide a source of recurrent excitation on SC neurons, amplifying the retinal drive (*Shi et al., 2017*). However, the cell-type specificity of the intracollicular amplification is still debated. For example, while GAD2-positive horizontal cells receive excitatory inputs from stellate cells (*Gale and Murphy, 2018*), intracollicular connectivity originating from neighboring excitatory cell types such as narrow-field (NF) and WF cells has shown to be rare (*Gale and Murphy, 2018*). Next-generation in vitro multi-neuron patch-clamp recordings (*Campagnola et al., 2022*) could reveal important new insights into the cell type-specific (*Ayupe et al., 2023*) intracollicular wiring.
4. In addition, collicular neurons in the visual layers also get inputs from deeper SC layers. However, excitatory connections from deep SC to the upper layers are rather uncommon (*Gale and Murphy, 2018*; *Ghitani et al., 2014*). Despite these studies have highlighted different sources of excitation onto inhibitory SC neurons, further studies are required that elucidate the role and strength of input that deep SC layers or external sources provide to drive inhibitory SC populations.

In this study, we recorded populations of well-isolated single units including RGCs as well as GABAergic and non-GABAergic SC neurons in VGAT-ChR2 mice in vivo. Our approach using VGAT-ChR2 mice to target all types of inhibitory SC neurons that express the vesicular GABA transporter VGAT enables a first impression on how INs sculpt retinocollicular signaling. Our inhibitory population is made up of 30% of the recorded SC neurons and connected pairs. These observations are in line with previous findings showing that in the upper SC layers, a substantial part of neurons (30%) are GABAergic (*Mize, 1992*; *Whyland et al., 2020*) and that around one third of the postsynaptic targets of retinocollicular terminals are GABAergic (*Boka et al., 2006*; *Whyland et al., 2020*).

In the last decade, the diversity of SC cell types has been mainly characterized based on morphology and response properties (*De Franceschi and Solomon, 2018*; *Ito et al., 2017*). Morphologically and functionally, four distinct classes of retinorecipient SC neurons have been identified: stellate, horizontal, WF, and NF neurons (*Gale and Murphy, 2016*; *Gale and Murphy, 2014*). Excitatory SC cell types such as WF, stellate, and NF cells differ in morphological characteristics. This morphological diversity might also be linked to how these cells integrate inputs from the retina. Smaller cells, such

as NF and stellate cells, typically have smaller RF sizes. Their limited dendritic arborization enables them to selectively respond to specific visual features. NF neurons show strong direction selectivity (DS) suggesting that these cells might be strongly driven by the retina, likely from DS-RGCs (*Gale and Murphy, 2014*). In contrast, WF cells have extensive dendritic arbors and large RF sizes. These cells likely integrate inputs from multiple RGC axons (*Gale and Murphy, 2014*).

The use of genetic mouse lines provides a valuable tool for understanding the circuitry and functional roles of different neuron classes by selectively targeting specific cell types. Advanced techniques have started to shed light on this diversity by providing cell-type characterizations on the molecular level (*Barchini et al., 2018*; *Gale and Murphy, 2018*; *Gale and Murphy, 2014*; *Li and Meister, 2022*; *Villalobos et al., 2018*). Three categories of potential GABAergic interneurons have been classified in the SC by the use of Cre-lines (*Whyland et al., 2020*) that can be further subdivided into intrinsic interneurons and potential projection neurons. A recent study by *Tsai et al., 2022*, has mapped retinocollicular connectivity by targeting specific RGC-SC circuits using a novel approach (Trans-Seq) to classify subsets of SC cell types that are innervated by genetically defined RGC types (*Tsai et al., 2022*). They identified five inhibitory clusters of retinorecipient SC cells that allow to specifically map retinocollicular connectivity. In addition, 12 distinct inhibitory subtypes across the entire SC circuit have been recently identified using single-nucleus RNA sequencing (*Ayupe et al., 2023*). Thus, we are at an enthralling stage where information on SC cell types finally becomes available based on gene expression patterns, alike to the cortical circuit (*Tremblay et al., 2016*). Likewise investigating how optogenetic manipulation of SC INs modulates functional properties of recorded SC EXNs (*Gale and Murphy, 2016*) is crucial for gaining a mechanistic understanding of visual processing within the SC circuitry. Hence, a key step forward would be to leverage the tracing and transcriptomics studies and combine these methods with our high-density recording approach to enable a finer characterization of the collicular interneuron diversity and the complexity of their connectivity patterns in vivo.

Taken together, our study targeted a heterogeneous population of GABAergic neurons in VGAT-ChR2 mice and our results reveal that EXNs and INs in the two main brain areas for visual processing, the visual cortex and the SC, integrate sensory afferent inputs in different ways.

## Materials and methods
### Experimental design
#### Animals
All experiments were carried out in accordance with the guidelines given by Landesamt für Gesundheit und Soziales (LAGeSo - G 0142/18) Berlin and were approved by the authority. Adult VGAT-ChR2 (Slc32a1-COP4∗H134R/EYFP, The Jackson Laboratory, stock no. 014548, n=9) transgenic mice of either sex were used to activate INs (*Zhao et al., 2011*). Mice were aged 7–9 months on the day of recordings.

#### Surgical procedures
All experiments were conducted in VGAT-ChR2 mice of both sexes. Mice were induced with 2.0–2.5% isoflurane (CP-Pharma) in an induction chamber and transferred to the surgery. Once anesthetized, the surgery was performed in a stereotactic frame (Narishige) with a closed-loop temperature controller (FHC DC) for monitoring the animal's body temperature and maintaining it at 37°C. The isoflurane level was gradually lowered during surgery and maintained at 0.8–1.5% while ensuring a complete absence of vibrissa twitching or responses to tactile stimulation. During surgery, the eyes were protected with eye ointment (Vidisic, Bausch + Lomb). Mice were head-fixed in the frame, and the skull was exposed. The head was aligned along the anterior-posterior (AP) axis and marks were made for the craniotomy at 600–1500 µm ML and 0–1500 µm AP from Lambda (Paxinos and Franklin, Nixdorf 2007 stereotactic atlas) using a micromanipulator (Luigs and Neumann). For optotagging experiments, an additional mark for the optic fiber placement was made at 3500 µm AP, 500–1200 µm ML to Bregma. A headpost was placed on the skull and implanted with dental cement (Paladur). Dental cement was also used to build a recording chamber to provide a bath for grounding. During this step, a silver wire (AG-10W, Science Products) was attached to the dental cement chamber for grounding. It is important to keep the dental cement chamber low enough on the anterior and posterior sides to allow for the shallow angle probe and fiber implantations, see *Sibille et al., 2022a*, for

a detailed description of the method (*Sibille et al., 2022b*). Craniotomies were made at the marked positions using a dental drill (Johnson-Promident). To allow for the probe insertion at a horizontal angle between 15° and 20°, a small part of the skull at the posterior part of the craniotomy was slightly thinned using the drill to ensure a smooth insertion (*Sibille et al., 2022b*).

## Anesthetized extracellular recordings and optogenetic stimulation

Electrophysiological recordings in SC were performed in nine anesthetized mice of both sexes (four female, five male) using Neuropixels 1.0 probes (*Jun et al., 2017*). Data were acquired at a sampling rate of 30 kHz using the Open Ephys acquisition software and GUI (*Siegle et al., 2017*) (https://www.open-ephys.org/) and the PXIe hardware acquisition system (National Instrument NI-PXIe-1071). The signal was amplified and stored in both the local field potential band (high pass-filtered 0–300 Hz) and the action potential band (300 Hz to 3 kHz). The tangential recordings along the AP axis in SC were performed as described in *Sibille et al., 2022a*. The fiber for optogenetic stimulation was inserted in the SC prior to the Neuropixels probe implantation to decrease mechanical pressure and reduce drift of the electrode probe. The angled optogenetic fiber was zeroed at the brain surface and inserted from the anterior in a 60–70° angle toward the AP axis (see *Figure 1A*) and slowly lowered using a manual micromanipulator until reaching an insertion of 150–200 µm. Following the fiber implantation, the Neuropixels probe was inserted tangentially in a 15–20° angle in the SC using a micromanipulator (NewScale). The probe was zeroed at Lambda and inserted 500–1200 µm ML and 2800–3800 µm DV. After the Neuropixels probe was implanted, the optic fiber was lowered further slowly (to a maximum insertion depth of ~250 µm) and LED pulses were applied to test for optogenetic activation of cells. After correct placement, the probe and fiber were allowed to settle for 15–20 min before data acquisition was started. In addition, confirmation of probe placement within the SC was done via RF mapping to check the number of channels with RFs. RFs were estimated from the multi-unit activity (MUA) using spike-triggered average (STA) (*Sibille et al., 2022b*; *Figure 1C*).

## Optotagging to identify GABAergic neurons in VGAT-ChR2 mice

We identified ('optotagged') populations of single GABAergic SC neurons via Channelrhodopsin-2 (ChR2) activation in VGAT-ChR2 mice. The optic fiber was calibrated prior to each experiment to ensure a stable light output (2.5 mW) measured at the fiber tip. Before the electrode probe insertion (see above), the optic fiber (NA.66, 200 µm core diameter, with a bare fiber tip, PlexBright, Plexon) coupled to a blue (470 nm) LED module (PlexBright, Plexon) was inserted using a 60–70° angle tilted toward the AP axis (*Figure 1A*). Blue light pulses were used to evoke spiking activity in VGAT-expressing neurons and to verify the proper alignment of the optic fiber and the Neuropixels probe. If no optogenetically evoked neuronal activity was observed, either the fiber was lowered to a maximum of 250 µm or the probe position was re-adjusted to ensure both the activation and recording of GABAergic SC neurons. Once the Neuropixels electrode probe and optic fiber were properly aligned, optogenetic stimulation using either a square wave or Gaussian function stimulus pulse was applied to identify light-activated channels (*Figure 1A and B*). As the stimulation using a square wave pulse led to the induction of light artifacts in the spiking response (*Jun et al., 2017*), we manually removed these artifacts in a post-processing step by interpolating the raw values in each channel during the onset and offset of the square pulse in 2 out of 11 experiments. In order to decrease light artifacts induced by the sharp light onset in the square wave pulse, we changed the pulse to using a Gaussian function stimulus pulse to gradually increase light intensity in the remaining nine experiments. The light pulses were applied during spontaneous conditions (presentation of a black background, 100 ms pulse duration, 2 Hz, 200 trials, 2.7–3.5 mW measured at the fiber tip) and conditions with visually evoked activity (checkerboard stimulus, 1000 ms pulse duration, 30 trials ON, 30 trials OFF, 2.7–3.5 mW measured at the fiber tip, n=10/11 experiments). Both stimulation protocols were shown twice within the recording to ensure stable responses throughout the recording session. GABAergic neurons were identified by their increased short-latency responses to blue light stimulation pulses. When we compared the visually evoked responses to those induced by ChR2 activation we found that they varied in their temporal activation profile (*Figure 1B and C*). The response to the visual sparse noise stimulus starts ~80 ms after the stimulus onset (*Figure 1C*) while responses induced by ChR2 activation had a shorter latency following light onset (*Figure 1C*).

## Visual stimulation

Visual stimuli were generated in Python using the PsychoPy2 toolbox (*Peirce et al., 2019*). The visual stimuli were displayed on a calibrated screen (Dell, refresh rate = 120 Hz, mean luminance = 120 cd/m²). For RF mapping, we presented the sparse noise visual stimulus on a grid of 24×14 squares. The sparse noise targets were either dark (on light background) or light (on dark background) and 10° in size with [n_targets per frame, n_trials per position] = [2,20] and presented for 100 ms. A set of standardized different visual stimuli was presented including a natural movie (30 trials, randomized with other stimuli, 10 s) taken from *Froudarakis et al., 2014*, and a full-field chirp stimulus (*Baden et al., 2016*). The frametimes of the visual stimuli were marked by stimulus-locked synchronizing events (TTLs).

## Histology

For histological reconstruction of the electrode track, the probe was removed, coated with fluorescent dye (DiI, Abcam-ab145311), and re-inserted in the same location. In a few experiments we performed multiple recordings and only stained the last insertion. Subsequently, the animal was terminated with an excess of isoflurane (>4%). Cardiac perfusion was performed with phosphate-bufferred saline (PBS) followed by 4% paraformaldehyde (PFA) in PBS. Brains were post-fixed overnight in 4% PFA, and stored in PBS until slicing (100 μm) on a vibratome (Leica VT1000S). The brain slices were mounted using a mounting medium containing DAPI (DAPI-Flouromount-G, Biozol Cat. 0100-20). Slices were imaged on a fluorescence microscope using a 2.5× objective for post hoc visualization of the electrode probe track.

## Data analysis and statistical analyses

Data analysis was performed in Python 3.0 (https://www.anaconda.com/) using custom-written scripts. MATLAB 2019b (https://www.mathworks.com/) was used for spike sorting, see below. Statistical tests were performed using the two-sided Wilcoxon rank-sum test for unpaired samples and the Wilcoxon signed-rank test (two-sided) for paired samples using the scipy.stats module in Python, unless stated otherwise. For correlation analyses we used Pearson correlation implemented via the linregress function. Population results are indicated as mean ± standard deviation if not stated otherwise. The number of neurons and animals used in each analysis is reported in the Results section or figure legends.

## MUA extraction

For MUA analysis, common average referencing was applied to the bandpass-filtered action potential data (Butterworth filter order 2, 0.3–3 kHz), where each event is extracted using custom-written Python scripts. Spike detection was performed for each channel independently at a threshold of four standard deviations in the action potential band (double side detection). The scripts for performing the MUA analysis are available on GitHub (*Sibille et al., 2022b*) (https://github.com/KremkowLab/tangential_recording; *Teh, 2023*).

## Offline spike sorting and criteria for unit classification

KiloSort2 and -3 (*Pachitariu et al., 2016*) (https://github.com/MouseLand/Kilosort, version 2 and 3; *Pachitariu, 2023*) were used for automatic detection and clustering of spikes followed by manual curation using phy2 (*Rossant et al., 2016*) (https://github.com/cortex-lab/phy, version 2.0; *Rossant, 2023*). Double-counted spikes were removed for each cluster (within ±0.16 ms) (*Siegle et al., 2021*). Furthermore, clusters with between-unit overlapping spikes that show above chance zero-lag peaks in the CCGs (peak windows ± 0.5 ms) were re-evaluated individually in phy to be either refined or removed. ISI violations were calculated as the ratio of the spikes within the refractory period (±1.5 ms) to the total number of spikes. Units with ISI >0.05% were removed. We also checked for stable firing of clusters throughout the recording. Furthermore, we excluded units from further analysis that were lying outside the optogenetic stimulation range (see below, 'Post hoc separation of GABAergic and non-GABAergic SC neurons', see *Figure 1B*) and outside the SC boundaries (from STA RFs on MUA, see *Figure 1C*).

## Waveform classification to identify RGC axons and SC neurons

A waveform classification approach was applied to distinguish action potentials from RGC axons and somatic action potentials from SC neurons (*Sibille et al., 2022a*). In brief, we calculated the multi-channel waveform (MCW) which reflects the spatiotemporal profile of the action potential signals. Afferent axons and somatic signals could be classified based on their distinct waveform which allowed us to classify clusters into afferent axon action potential vs somatic action potential. In essence, due to the SC electrode probe implantation along the AP axis, the electrode captures signals from retinal axons that innervate the SC along the electrode channels. This propagation of RGC axonal action potentials along the axonal path is visible in the spatiotemporal profile of their MCW (*Figure 1D*, left).

## Waveform classification approach to separate GABAergic and non-GABAergic SC neurons

We performed standard waveform analyses to attempt to separate GABAergic SC neurons based on waveform features (*Jia et al., 2019*; *Quirk et al., 2009*; *Figure 1—figure supplement 2*). Briefly, we detected the negative (trough) and positive peak (peak) from single-channel waveforms and calculated their peak-to-trough duration. The amplitude was defined as the absolute difference between trough and peak. The peak-to-trough ratio was defined as the ratio between amplitudes of peak and trough.

## Post hoc separation of GABAergic and non-GABAergic SC neurons

To study retinal input integration in a cell type-specific manner, we used a method to separate the neural responses of GABAergic neurons from that of EXNs in the SC. First, we identified the minimum and maximum recording sites on the probe that were photo-activated. These borders were defined via the minimum and maximum peak channel of light-induced units that are activated on the probe. SC neurons that did not fall inside this optogenetic stimulation range (identified by their peak channel) were excluded from further analysis. We employed stringent criteria for determining the boundaries of LED-evoked response and selection of neurons to be included (*Figure 1—figure supplement 1*). The distribution of excitatory and inhibitory SC neurons is not significantly different for 9/11 experiments (Wilcoxon rank-sum, p-values = 0.307, 0.0115, 0.7.55, 0.834, $5.01\times10^{-6}$, 0.79, 0.80, 0.26, 0.33, 0.08, 0.13). In the two significantly different experiments, only two RGC-SC EXN pairs were located in the region without identifying SC INs, and thus will not affect the results. SC neurons were separated based on their short-latency increase in responses to blue light pulses during two different protocols: (1) baseline activity (black background, 100 ms pulse duration, 2 Hz pulse frequency) and (2) during a checkerboard stimulus (1000 ms pulse duration, 30 trials LED ON, 30 trials LED OFF). This classification was done using a custom-written GUI approach where units were labeled as either 'light-responsive', 'non light-responsive'. The labeling step was conducted by three independent observers. A unit was included if two out of three observers chose the same label. Units with unstable firing throughout the recording period were excluded from further analysis.

## Identification of synaptically connected RGC-SC pairs

Monosynaptic connections between RGC axons and SC neurons were detected using established methods (*Bereshpolova et al., 2020*; *Clay Reid and Alonso, 1995*; *Denman and Contreras, 2015*; *Usrey et al., 1998*) based on statistically significant peaks at synaptic delays. Briefly, we calculated a jittered (timescale jitter 5 ms) version of each spike train by randomizing all spike times within a consecutive 10–15 ms window (*Smith and Kohn, 2008*). We then calculated the cross-correlation between a pair of neurons both for the original (raw CCG) and the jittered spike train (jittered CCG). Subtracting the jittered CCG from the raw CCG results in a jitter-corrected version of the CCG which was used for monosynaptic pairs detection. Connected pairs were identified using cross-correlation analysis and significant peaks (peak detection window from +0.5 to 3.5 ms) had to extend over the threshold (baseline + 4x standard deviation) for at least five consecutive time bins (0.1 ms resolution) (as described previously, in *Sibille et al., 2022a*). The cross-correlations were calculated using the Python pycorrelate package (https://github.com/tritemio/pycorrelate; *Ingargiola, 2017*). Please note that for CCG analysis, spike trains were correlated over the full recording duration, except for

periods with optogenetic stimulation. Latencies to peak response were calculated from the CCGs as the maximum time within the response window.

## Analysis of retinocollicular connection strength

Synaptic efficacy and contribution measures of connected RGC-SC pairs were estimated from spike trains during the entire recording session (except for optotagging periods which were excluded) using standard approaches (*Bereshpolova et al., 2020*; *Clay Reid and Alonso, 1995*; *Swadlow and Gusev, 2002*; *Usrey et al., 1998*). Briefly, efficacy was estimated from the baseline-corrected CCGs by dividing the area of the CCG peak (baseline window: > –1 to <0.4 ms, peak window: >0.4 to <3.5 ms) by the total number of presynaptic spikes. Thus, an efficacy measure of 1 (100%) would reflect that for each presynaptic retinal spike, a postsynaptic spike could be detected. To estimate the connection contribution, we counted the number of SC spikes that were preceded by a retinal afferent spike, integrated over a time window between –3 and –0.5 ms, and divided this number by the total number of SC spikes. A contribution of 1 (100%) indicates that all spikes of an SC neuron are preceded by retinal afferent spikes.

## Paired-spike dynamics analysis

For estimation of paired-spike dynamics, baseline-corrected CCGs were re-calculated triggered on first and second RGCs using the same baseline and peak windows as mentioned above. We identified pairs of retinal spikes (first and second RGC) for each RGC-SC connected pair (*Usrey et al., 1998*). Briefly, a pair of retinal spikes was defined by a specific ISI where the second RGC spike followed the first RGC by an ISI of 5–30 ms. The second criterion for classifying a pair of retinal spikes was a certain dead time where the first retinal spike was preceded by a minimum period of silence (dead time: 30 ms). The dead time criterion was applied to ensure a comparable level of activity immediately preceding all first spikes. The preceding window (>–1 to <0.4 ms) was used as a baseline window. Paired-spike enhancement was calculated as the ratio of efficacy for the second RGC to the efficacy for the first RGC.

## Functional similarity analysis of retinocollicular connected pairs

To characterize the functional similarity between connected RGCs and the target neurons in the SC, we estimated the functional similarity index. To do so, we calculated the correlation coefficients from responses during the light and dark sparse noise ($r_{SL}$ and $r_{SD}$) as well as during the natural movie stimulus ($r_{NM}$), phase scrambled movie ($r_{PSM}$), and during a checkerboard stimulus ($r_{Chk}$). The similarity index was calculated between the responses of the RGC and the responses of the postsynaptic SC neuron. As a result, the similarity index reflects the degree of similarity in the responses of the connected pairs. The correlation of activity during sparse noise stimuli ($r_{SL}$ and $r_{SD}$) was estimated from the spatio-temporal receptive fields (STRFs) which were calculated from the STA. The similarity during the other stimuli ($r_{NM}$, $r_{PSM}$ and $r_{Chk}$) were calculated from the PSTHs. We limited this analysis step to neurons with high signal-to-noise ratio (SNR) in the visually evoked activity (SNR >8 for the STRFs). The overall similarity index was then calculated from the averaged correlation coefficients, functional similarity index = $(r_{SD} + r_{SL} + r_{NM} + r_{PSM} + r_{Chk})/5$.

# Acknowledgements

We thank J Poulet for providing us the VGAT-ChR2 mice; KL Teh, T Lupashina, and J Kosubek-Langer for comments on the manuscript. We thank the Neuropixels community for their support. This work was supported by the DFG Emmy-Noether grants KR 4062/4-1 and KR 4062/4-2 (JK).

# Additional information

## Funding

| Funder | Grant reference number | Author |
|---|---|---|
| Deutsche Forschungsgemeinschaft | KR 4062/4-1 | Jens Kremkow |
| Deutsche Forschungsgemeinschaft | KR 4062/4-2 | Jens Kremkow |

The funders had no role in study design, data collection and interpretation, or the decision to submit the work for publication.

## Author contributions

Carolin Gehr, Conceptualization, Data curation, Formal analysis, Investigation, Visualization, Methodology, Writing - original draft, Writing - review and editing; Jeremie Sibille, Conceptualization, Supervision, Methodology, Project administration; Jens Kremkow, Conceptualization, Resources, Supervision, Funding acquisition, Validation, Writing - original draft, Project administration, Writing - review and editing

## Author ORCIDs

Carolin Gehr ⓘ http://orcid.org/0000-0003-0892-506X
Jeremie Sibille ⓘ http://orcid.org/0000-0001-6895-7405
Jens Kremkow ⓘ http://orcid.org/0000-0001-7077-4528

## Ethics

This study was performed in strict accordance with the guidelines given by Landesamt fur Gesundheit und Soziales (LAGeSo - G 0142/18) Berlin and were approved by the authority. Every effort was made to minimize suffering.

Reviewer #1 (Public Review): https://doi.org/10.7554/eLife.88289.3.sa1
Reviewer #2 (Public Review): https://doi.org/10.7554/eLife.88289.3.sa2
Reviewer #3 (Public Review): https://doi.org/10.7554/eLife.88289.3.sa3
Author Response https://doi.org/10.7554/eLife.88289.3.sa4

# Additional files

## Supplementary files

• MDAR checklist

## Data availability

Data supporting the findings of this study presented in the figures are available in Figure 1—source data 1, Figure 2—source data 1, Figure 3—source data 1, Figure 4—source data 1, and Figure 5—source data 1. Python code of the analysis is available in Figure 1—source code 1, Figure 2—source code 1, Figure 3—source code 1, Figure 4—source code 1 and Figure 5—source code 1. Data and Python code are available in a public GitHub repository (https://github.com/KremkowLab/Gehr_et_al_2023_eLife, copy archived at *KremkowLab, 2023*).

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
