## [Editor Report · eLife assessment]

This **valuable** study contributes to understanding how retinal activity shapes the response properties of excitatory and inhibitory neurons in a major visual target, the superior colliculus. The evidence supporting the claim is **convincing**: the work is technically excellent yet the interpretation of these results assumes an unbiased sampling and integration of the RGC axon in the SC, a caveat pointed out by the authors. Overall, this study provides insights into the integration of visual information from the eye to the brain, and this work will be of interest to visual neuroscientists.

---

## [Referee Report · Reviewer #1 (Public Review)]

Gehr and colleagues used an elegant method, using neuropixels probes, to study retinal input integration by mouse superior collicular cells in vivo. Compared to a previous report of the same group, they opto-tagged inhibitory neurons and defined the differential integration onto each group. Through these experiments, the author concluded that overall, there is no clear difference between the retina connectivity to excitatory and inhibitory superior colliculus neurons. The exception to that rule is that excitatory neurons might be driven slightly stronger than inhibitory ones. Technically, this work is performed at a high level, and the plots are beautifully conceived, but I have doubts if the interpretation given by the authors is solid. I will elaborate below.

Some thoughts about the interpretation of the results.

My main concern is the "survivor bias" of this work, which can lead to skewed conclusions. From the data set acquired, 305 connections were measured, 1/3 inhibitory and 2/3 excitatory. These connections arise from 83 RGC onto 124 RGC (I'm interpreting the axis of Fig.2 C). Here it is worth mentioning that different RGC types have different axonal diameters (Perge et al., 2009). Here the diameter is also related to the way cells relay information (max frequencies, for example). It is possible that thicker axons are easier to measure, given the larger potential changes would likely occur, and thus, selectively being picked up by the neuropixels probe. If this is the case, we would have a clear case of "survival bias", which should be tested and discussed. One way to determine if the response properties of axonal termini are from an unbiased sample is to make a rough functional characterization as generally performed (see Baden et al. 2006). This is fundamental since all other conclusions are based on unbiased sampling.

One aspect that is not clear to me is to measure of connectivity strength in Figure 2. Here it seems that connectivity strength is directly correlated with the baseline firing rate of the SC neuron (see example plots). If this is a general case, the synaptic strength can be assumed but would only differ in strength due to the excitability of the postsynaptic cell. This should be tested by plotting the correlation coefficient analysis against the baseline firing rate.

My third concern is the assessment of functional similarity in Fig. 3. It is not clear to me why the similarity value was taken by the arithmetic mean. For example, even if the responses are identical for one connected pair that exclusively responds either to the ON or OFF sparse noise, the maximal value can only be 0.67. Perhaps I misunderstood something. Secondly, correlations in natural(istic) movies can differ dramatically depending on the frame rate that the movie was acquired and the way it is displayed to the animal. What looks natural to us will elicit several artifacts at a retinal level, e.g., due to big jumps between frames (no direction-selective response) or overall little modulation (large spatial correlations). I would rather opt for uniform stimuli, as suggested previously. Of course, these are also approximations but can be easily reproduced by different labs and are not subjected to the intricacies of the detailed naturalistic stimulus used.

Fourth. It is important to control the proportion of inhibitory cells activated optogenetically across the recording probe. Currently, it is not possible to assess if there are false negatives. One way of controlling for this would be to show that the number of inhibitory interneurons doesn't vary across the probe.

Fifth. In Fig. 4, the ISI had a minimal bound of 5 ms. Why? This would cap the firing rate at 200Hz, but we know that RGC in explants can fire at higher frequencies for evoked responses. I would set a lower bound since it should come naturally from the after-depolarization block. Another aspect that remains unclear is to what extent the paired-spike ratio depends on the baseline firing rate. This would change the interpretation from the particular synaptic connection to the intrinsic properties of the cell and is plausible since the bassline firing rate varies tremendously. One related analysis would be to plot the change of PSR depending on the ISI. It would be intuitive to make a scatter plot for all paired spikes of all recorded neurons (separated into inhibitory and excitatory) of ISI vs. PSR.

Panel 4E is confusing to me. Here what is plotted is efficacy 1st against PSR (which is efficacy 2nd/efficacy 1st). Given that you have a linear relation between efficacy 1st and efficacy 2nd (panel 4C), you are essentially re-plotting the same information, which should necessarily have a hyperbolic relationship: [ f(x) = y/x ]. Thus, fitting this with a linear function makes no sense and it has to be decaying if efficacy 2nd > efficacy1st as shown in 4C.

Finally, in Figure 5, the perspective is inverted, and the spike correlations are seen from the perspective of SC neurons. Here it would also be good to plot the cumulative histograms and not look at the averages. Regarding the similarity index and use of natural stats, please see my previous comments. Also, would it be possible to plot the contribution v/s the firing rate with the baseline firing rate with no stimulation or full-field stimulation? This is important since naturalistic movies have too many correlations and dependencies that make this plot difficult to interpret.

Overall, the paper only speaks from excitatory and inhibitory differences in the introduction and results. However, it is known that there are three clear morphologically distinct classes of excitatory neurons (wide-field, narrow-field, and stellate). This topic is touched in the discussion but not directly in the context of these results. Smaller cells might likely be driven much stronger. Wide-field cells would likely not be driven by one RGC input only and will probably integrate from many more cells than 6.

---

## [Referee Report · Reviewer #2 (Public Review)]

This study follows up on a previous study by the group (Sibille et al Nature Communications 2022) in which high density Neuropixel probes were inserted tangentially through the superficial layers of the superior colliculus (SC) to record the activity of retinocollicular axons and postsynaptic collicular neurons in anesthetized mice. By correlating spike patterns, connected pairs could be identified which allowed the authors to demonstrate that functionally similar retinal axon-SC neuron pairs were strongly connected.

In the current study, the authors use similar techniques in vGAT-ChR2 mice and add a fiber optic to identify light-activated GABAergic and non-light-activated nonGABAergic neurons. Using their previously verified techniques to identify connected pairs, within regions of optogenetic activation they identified 214 connected pairs of retinal axons and nonGABAergic neurons and 91 pairs of connected retinal axons and GABAergic neurons. The main conclusion is that retinal activity contributed more to the activity of postsynaptic nonGABAergic SC neurons than to the activity of postsynaptic GABAergic SC neurons.

The study is very well done. The figures are well laid out and clearly establish the conclusions. My main comments are related to the comparison to other circuits and further questions that might be addressed in the SC.

It is stated several times that the superior colliculus and the visual cortex are the two major brain areas for visual processing and these areas are compared throughout the manuscript. However, since both the dorsal lateral geniculate nucleus (dLGN) and SC include similar synaptic motifs, including triadic arrangements of retinal boutons with GABAergic and nonGABAergic neurons, it might be more relevant to compare and contrast retinal convergence and other features in these structures.

The GABAergic and nonGABAergic neurons showed a wide range of firing rates. It might be interesting to sort the cells by firing rates to see if they exhibit different properties. For example, since the SC contains both GABAergic interneurons and projection neurons it would be interesting to examine whether GABAergic neurons with higher firing rates exhibit narrower spikes, similar to cortical fast spiking interneurons. Similarly, it might be of interest to sort the neurons by their receptive field sizes since this is associated with different SC neuron types.

The recording techniques allowed for the identification of the distance between connected retinocollicular fibers and postsynaptic neurons. It might also be interesting to compare the properties of connected pairs recorded at dorsal versus ventral locations since neurons with different genetic identities and response properties are located in different dorsal/ventral locations (e.g. Liu et al. Neuron 2023). Also, regarding the strength of connections, previous electron microscopy studies have shown that the retinocollicular terminals differ in density and size in the dorsal/ventral dimension (e.g Carter et al JCN 1991).

Was optogenetic activation of GABAergic neurons ever paired with visual activation? It would be interesting to examine the receptive fields of the nonGABAergic neurons before and after activation of the GABAergic neurons (as in Gale and Murphy J Neurosci 2016).

---

## [Referee Report · Reviewer #3 (Public Review)]

This study performs in vivo recordings of neurons in the mouse superior colliculus and their afferents from the retina, retinal ganglion cells (RGCs). Building on a preparation they previously published, this study adds the use of optogenetic identification of inhibitory neurons (aka optotagging) to compare RGC connectivity to excitatory and inhibitory neurons in SC. Using this approach, the authors characterize connection probability, strength, and response correlation between RGCs and their target neurons in SC, finding several differences from what is observed in the retina-thalamus-visual cortex pathway. As such, this may be a useful dataset for efforts to understand retinocollicular connectivity and computations.

---

## [Author Response]

The following is the authors’ response to the original reviews.

We are grateful for the comments from the reviewers, which helped us to strengthen our analyses and communicate more effectively the details of our findings and their significance. To address their criticisms, we have performed new analyses and revised the text and figures. We believe the manuscript was significantly improved. We provide the line number of important parts of the text that were changed, here in this letter. Below, we address the specific comments from the reviewers in detail.

**Reviewer #1 (Public Review):**
Gehr and colleagues used an elegant method, using neuropixels probes, to study retinal input integration by mouse superior collicular cells in vivo. Compared to a previous report of the same group, they opto-tagged inhibitory neurons and defined the differential integration onto each group. Through these experiments, the author concluded that overall, there is no clear difference between the retina connectivity to excitatory and inhibitory superior colliculus neurons. The exception to that rule is that excitatory neurons might be driven slightly stronger than inhibitory ones. Technically, this work is performed at a high level, and the plots are beautifully conceived, but I have doubts if the interpretation given by the authors is solid. I will elaborate below.Some thoughts about the interpretation of the results.My main concern is the "survivor bias" of this work, which can lead to skewed conclusions. From the data set acquired, 305 connections were measured, 1/3 inhibitory and 2/3 excitatory. These connections arise from 83 RGC onto 124 RGC (I'm interpreting the axis of Fig.2 C). Here it is worth mentioning that different RGC types have different axonal diameters (Perge et al., 2009). Here the diameter is also related to the way cells relay information (max frequencies, for example). It is possible that thicker axons are easier to measure, given the larger potential changes would likely occur, and thus, selectively being picked up by the neuropixels probe. If this is the case, we would have a clear case of "survival bias", which should be tested and discussed. One way to determine if the response properties of axonal termini are from an unbiased sample is to make a rough functional characterization as generally performed (see Baden et al. 2006). This is fundamental since all other conclusions are based on unbiased sampling.

First of all, we want to thank the reviewer for the detailed and constructive comments based on which we refined the analysis and updated the figures. We hope that our changes adequately address the concerns of reviewer #1.

We would like to clarify that Fig. 2C represents an example from a single experiment. In total, we recorded 326 RGCs and 680 SC neurons in total, with 161 individual RGCs making connections onto 183 individual SC neurons. Moreover, we thank the reviewer for bringing up that important point about the potential “survivor bias”. To address this concern, we would like to provide some clarifications (see below). In addition, we now added the point that different RGCs can have different axonal diameter as requested by the reviewers (line 605).

It is important to note that our approach does not capture the total pool of retinal inputs. Moreover, we did not want to convey the impression that our approach equally captures all retinal inputs to a given SC neuron, as this is not the case. Likewise, it is important to note that our current method does not allow for the measurement of axonal diameters. To obtain an estimate of axonal thickness, complementary techniques such as imaging/staining or electron microscopy would be needed. Our study aimed at characterizing connected RGC-SC pairs and how excitatory and inhibitory neurons in the SC integrate retinal inputs, providing valuable insight on their wiring principles.

We greatly appreciate the reviewer for highlighting this limitation and we now address these points in the discussion of the revised version of our manuscript (line 603).

Regarding the suggested “rough functional characterization” of the RGCs. We have thought about this analysis and unfortunately, we did not present the necessary stimuli, e.g. chirp, in all experiments to be able to perform this analysis. Moreover, the dataset represented in this work contains only 326 RGC neurons, with 161 identified RGCs making connections to SC neurons. Thus, it is unlikely that our dataset uniformly covers all ~30 RGC types in the mouse. However, given that our dataset is the first measurement of RGC inputs to SC INs and SC EXNs in vivo, we believe it provides a first step and a foundation for future studies focusing on specific RGC types to refine our understanding of the RGC-SC circuitry. We discuss this point now in the revised manuscript (line 586).

One aspect that is not clear to me is to measure of connectivity strength in Figure 2. Here it seems that connectivity strength is directly correlated with the baseline firing rate of the SC neuron (see example plots). If this is a general case, the synaptic strength can be assumed but would only differ in strength due to the excitability of the postsynaptic cell. This should be tested by plotting the correlation coefficient analysis against the baseline firing rate.

We appreciate the reviewer for bringing up this important point. From the analysis perspective, we would like to clarify that the efficacy measure is independent of the baseline firing rate. It quantifies the probability of adding spikes on top of the baseline rate by subtracting the baseline firing rate before measuring the area of the peak (Usrey et al., 1999).

Furthermore, we acknowledge the reviewer’s interesting and valuable observation about the relationship of the firing rate and the excitability of the SC neuron in the example plots. To test whether the efficacy is directly related to the mean firing rate, we conducted additional analyses to show the efficacy measure as a function of the mean firing rate (Author response image 1 and Figure 2G). To that end, we utilized two different measures of firing rate: the mean firing rate during spontaneous activity (gray screen) over a duration of 10 sec (across 30 trials), which was interleaved with the natural movie presentations, and the overall firing rate throughout the entire recording session. Our findings indeed reveal a positive correlation, as predicted by the reviewer (Author response image 1, gray screen: EXC r = 0.22721; p < 0.00081; INH: r = 0.34677, p = 0.00076; entire recording: EXC r = 0.42685; p < 0.0005; INH: r = 0.43543, p = 0.00002).

**Author response image 1. sa4fig1:** Efficacy measure of connected RGC-SC pairs as a function of the mean firing rate during different stimulus conditions: during spontaneous activity (gray screen, left) and throughout the entire recording session (right).

However, it is important to note that although we observe a correlation on the population level, the relationship between postsynaptic firing and efficacy is diverse. We identify pairs with strong connections despite the firing rate of the postsynaptic SC cell being low. Likewise, we also find pairs with weak connections despite the firing rate of the SC neuron being high (Author response image 2). These observations suggest that factors beyond the postsynaptic firing contribute to the efficacy of the connection. This is exemplified by the fact that SC neurons can receive both strong and weak connections from their convergent presynaptic RGC pool.

**Author response image 2. sa4fig2:** RGC-SC connectivity. Cross-correlograms showing 4 connected RGC-SC pairs (top) with two RGCs connecting onto the same SC neuron. Raster plots of SC neuron spiking activity in response to firing of the presynaptically connected RGC. The same SC neuron can receive both strong and weak RGC inputs.

In summary, we thank the reviewer for bringing up this important question, and we believe that our additional analyses shed light on the relationship between firing rate and efficacy. This result is very interesting, and we include these findings in the updated Figure 2 in the revised manuscript (panel 2G) in exchange with the panel of the peak latency. Moreover, we also address this point now in the results and discussion section of the revised manuscript (line 280 and line 525).

My third concern is the assessment of functional similarity in Fig. 3. It is not clear to me why the similarity value was taken by the arithmetic mean. For example, even if the responses are identical for one connected pair that exclusively responds either to the ON or OFF sparse noise, the maximal value can only be 0.67. Perhaps I misunderstood something.

We thank the reviewer for raising this point about the clarification regarding the calculation of the similarity index. We apologize for any confusion caused by our description on the similarity index calculation. To clarify, the similarity index was calculated specifically between the responses of the RGC and the responses of the postsynaptic SC neuron, rather than between the neurons and the visual stimulus. As a result, the similarity index reflects the degree of similarity in the responses of the connected pairs. Therefore, if the responses of the RGC and the connected postsynaptic neuron are identical, regardless of whether they respond exclusively to ON, only to OFF, or a mixture of ON-OFF, the similarity index will be one. We have updated the relevant part in the methods section to make this point clearer to the reader (line 917).

Secondly, correlations in natural(istic) movies can differ dramatically depending on the frame rate that the movie was acquired and the way it is displayed to the animal. What looks natural to us will elicit several artifacts at a retinal level, e.g., due to big jumps between frames (no direction-selective response) or overall little modulation (large spatial correlations). I would rather opt for uniform stimuli, as suggested previously. Of course, these are also approximations but can be easily reproduced by different labs and are not subjected to the intricacies of the detailed naturalistic stimulus used.

We agree with the reviewer that spatiotemporal correlations of naturalistic stimuli are complex. To address this point, we added two stimuli with little spatiotemporal correlations to the similarity analysis. The first stimulus we added is a phase scrambled version of the natural movie (PSM, also taken from Froudarakis et al. (2014)). The second is a binary white noise checkerboard stimulus. These stimuli were presented randomly interleaved with the natural movie, for 30 trials each. The similarity index analysis revealed that even with uniform stimuli included, the average similarity index is correlated to the efficacy. We show this data now in Figure 3.

Fourth. It is important to control the proportion of inhibitory cells activated optogenetically across the recording probe. Currently, it is not possible to assess if there are false negatives. One way of controlling for this would be to show that the number of inhibitory interneurons doesn't vary across the probe.

We thank the reviewer for highlighting this important aspect of the experiment and analysis. We are aware of this point and therefore took extra care to minimize the biases that could be introduced by our recording and stimulation method. Our approach to include recorded excitatory and inhibitory neurons was conservative, briefly:

1. We included only excitatory and inhibitory neurons that were within the SC, defined by visually driven activity and continuous retinotopy (see method).

2. We further restricted the included neurons to neurons that were located within the boundaries of the LED evoked responses, i.e. the recording channels with optogenetic evoked MUA responses within the SC (Figure 1 – figure supplement 1).

3. Both excitatory and inhibitory SC neurons were selected in this way.

These inclusion criteria were specifically designed to avoid sampling excitatory neurons from regions on the Neuropixels probe that lacked optogenetically evoked responses and thus to minimize the number of falsely labeled excitatory neurons.

To illustrate these inclusion criteria and the resulting spatial distribution of the selected excitatory and inhibitory SC neurons along the 384 channels of the Neuropixels probe, we now added a supplementary figure (Figure 1 – figure supplement 1). This figure shows the multi- unit activity in response to optogenetic stimulation and the distribution of inhibitory and excitatory single units within the range of channels that are activated via LED stimulation for 3/11 selected experiments. This highlights that we employed stringent criteria for determining the boundaries and selecting which neurons to include in our study. The distribution of excitatory and inhibitory SC neurons is not significantly different for 9/11 experiments (Wilcoxon rank-sum test, p values = 0.307, 0.0115, 0.755, 0.834, 5.01×10-6, 0.79, 0.80, 0.26, 0.33, 0.08, 0.13). Moreover, in the two significantly different experiments only 2 RGC-SC EXC pairs were located in the region without identified SC INs, and thus will not affect the results. We now address this point in the methods section (line 859).

Fifth. In Fig. 4, the ISI had a minimal bound of 5 ms. Why? This would cap the firing rate at 200Hz, but we know that RGC in explants can fire at higher frequencies for evoked responses. I would set a lower bound since it should come naturally from the after-depolarization block.

The chosen 5 ms minimal bound was in the range used in previous literature, e.g. 4-30 ms in Usrey et al. 1998 (Usrey et al., 1998). To address the question of the reviewer, we re-analyzed the data with a lower bound of 2 ms (2 – 30 ms) to include RGCs that fire at higher frequencies than 200Hz. However, we did not observe a clear difference between the 2-30 and 5-30 ms groups for inhibitory connections (SC IN: p = 0.604). Only the excitatory connections show a statistically significant difference (p = 0.011), however, the effect size is small (Cohen’s d = EXC = 0.063, INH = 0.030). Nonetheless we updated a panel in figure 4 to represent the 2-30 ms group (Figure 4F).

Another aspect that remains unclear is to what extent the paired-spike ratio depends on the baseline firing rate. This would change the interpretation from the particular synaptic connection to the intrinsic properties of the cell and is plausible since the bassline firing rate varies tremendously.

To address how the paired-spike ratio depends on the baseline firing rate we plotted the change of PSR depending on ISI as suggested by the reviewer.

One related analysis would be to plot the change of PSR depending on the ISI. It would be intuitive to make a scatter plot for all paired spikes of all recorded neurons (separated into inhibitory and excitatory) of ISI vs. PSR.

We appreciate the valuable suggestion from the reviewer. We have now separated the ISIs into distinct groups spanning 5 ms intervals represented in Author response image 3, right. These intervals range from 5-10 ms up to 25-30 ms. Notably, we observe a difference between the excitatory and inhibitory populations. The excitatory population exhibits a monotonic decrease in mean PSR across the intervals, while the inhibitory population shows a peak around 10/15 ms.

**Author response image 3. sa4fig3:** Change of mean paired-spike ratio (PSR) depending on ISI. (Left) Comparison of PSR between two groups of different ISIs. The 2-30 ms group ensures to include high-firing RGCs (excitatory pairs 2-30 vs 5-30 ms p = 0.011; inhibitory pairs 2-30 vs 5-30 ms p = 0.604, Wilcoxon signed-rank). (Right) PSR for groups of different ISI intervals. Mean PSR ± SEM for excitatory groups: 2.0±0.09, 1.75±0.09, 1.51±0.05, 1.31±0.05, 1.2±0.05; inhibitory groups: 1.35±0.06, 1.51±0.09, 1.5±0.1,1.22±0.06, 1.21±0.07. p E vs I (within group): 1.55±10-5, 9.55±10-2, 4.21±10-1, 3.74±10-1, 6.22 ±10-1, Wilcoxon rank-sum test.

Panel 4E is confusing to me. Here what is plotted is efficacy 1st against PSR (which is efficacy 2nd/efficacy 1st). Given that you have a linear relation between efficacy 1st and efficacy 2nd (panel 4C), you are essentially re-plotting the same information, which should necessarily have a hyperbolic relationship: [ f(x) = y/x ]. Thus, fitting this with a linear function makes no sense and it has to be decaying if efficacy 2nd > efficacy1st as shown in 4C.

We thank the reviewer for raising this question which helped us to improve the representation and disruption of the results shown in figure 4. Panel 4E is intended to investigate whether there is a correlation between the efficacy strength (eff 1st) and the amount of facilitation (PSR). From panel 4C it is already evident that the data points for high efficacies lie closer to the unity line, as compared to the data points for low efficacies. This suggests that the PSR is stronger for connections with smaller efficacies 1st. To quantify this relationship, we have plotted the efficacy 1st vs the PSR in panel 4E, which thus adds new information to the figure. Importantly, this panel is shown in log-log scales, and therefore the decaying relationship is not evident. If we had shown the data on linear-linear scale, the decaying function would have been evident (Author response image 4). And indeed, as the reviewer pointed out, we cannot fit a hyperbolic relationship with a linear function. This is exactly the reason why we show the data in log-log scale and also estimate the Pearson correlation also from the logs of the efficacies and PSRs.

In Author response image 4 we show the relationship plotted on linear scale using an approach to fit the hyperbolic relationship employing a hyperbolic cosecant function a/sinh⁡(b∗x)+c.

**Author response image 4. sa4fig4:** Relationship between efficacy to 1st RGC and PSR visualized on linear scale using a hyperbolic fitting approach a/sinh⁡(b∗x)+c.

Finally, in Figure 5, the perspective is inverted, and the spike correlations are seen from the perspective of SC neurons. Here it would also be good to plot the cumulative histograms and not look at the averages.

We added the cumulative histogram in Figure 5 (panel B), in addition to represent the raw data points and the mean.

Regarding the similarity index and use of natural stats, please see my previous comments. Also, would it be possible to plot the contribution v/s the firing rate with the baseline firing rate with no stimulation or full-field stimulation? This is important since naturalistic movies have too many correlations and dependencies that make this plot difficult to interpret.

We now show the contribution vs firing rates for different stimulus conditions in a new figure supplement (Figure 5- figure supplement 1). We added the correlations to the different stimuli for baseline firing rate with no stimulation (gray background), full-field stimulation (checkerboard) and phase scrambled natural movie.

Overall, the paper only speaks from excitatory and inhibitory differences in the introduction and results. However, it is known that there are three clear morphologically distinct classes of excitatory neurons (wide-field, narrow-field, and stellate). This topic is touched in the discussion but not directly in the context of these results. Smaller cells might likely be driven much stronger. Wide-field cells would likely not be driven by one RGC input only and will probably integrate from many more cells than 6.

We thank the reviewer for this comment. We agree with the reviewer that addressing how the different excitatory and inhibitory cell-types integrate RGC input is important to understand the visual processing mechanisms in the SC. The presented study aimed at comparing the excitatory and inhibitory population in general using the VGAT-ChR2 mouse line. Understanding how specific genetically defined cell-types integrate RGC inputs is clearly very interesting and should be done. Unfortunately, the mouse lines that would allow targeting genetically identified inhibitory cell-types are still limited and therefore we can only use functional measurements to assess different types of neurons in the SC. We now address this point about distinct SC cell-types in the discussion (line 643).

One possible functional measurement is the size of the receptive field, which, to some degree, could be used as a proxy for different morphologies, i.e. small receptive fields could hint towards compact morphology while large receptive fields could indicate a wider morphology. It is known for example that narrow-field and stellate cells have small RF sizes, while wide-field cells have large RFs. We studied the relationship between the RF size and spike waveform duration but did not find a significant correlation (Figure R6). Moreover, the spike waveform duration, as discussed in the manuscript, is not a valid criterion to separate EXNs and INs in the SC, as it is common practice in the cortex. We now also looked into whether the connectivity strength is related to the RF size. Interestingly, while in the current dataset we do not find a significant correlation between the efficacy and the receptive field size for both EXN and IN (Author response image 5, left), we do find a significant negative correlation between contribution and receptive field size for the excitatory neurons (Author response image 5, right). This result indicates that SC excitatory neurons with small receptive fields are more strongly coupled to the RGC input as compared to neurons with larger receptive fields.

**Author response image 5. sa4fig5:** Relationship between RF size and connectivity measures (efficacy and contribution) for RGC-SC EXN and RGC-SC IN pairs (two-sided Wilcoxon rank-sum test).

**Reviewer #2 (Public Review):**
This study follows up on a previous study by the group (Sibille et al Nature Communications 2022) in which high density Neuropixel probes were inserted tangentially through the superficial layers of the superior colliculus (SC) to record the activity of retinocollicular axons and postsynaptic collicular neurons in anesthetized mice. By correlating spike patterns, connected pairs could be identified which allowed the authors to demonstrate that functionally similar retinal axon-SC neuron pairs were strongly connected.In the current study, the authors use similar techniques in vGAT-ChR2 mice and add a fiber optic to identify light-activated GABAergic and non-light-activated nonGABAergic neurons. Using their previously verified techniques to identify connected pairs, within regions of optogenetic activation they identified 214 connected pairs of retinal axons and nonGABAergic neurons and 91 pairs of connected retinal axons and GABAergic neurons. The main conclusion is that retinal activity contributed more to the activity of postsynaptic nonGABAergic SC neurons than to the activity of postsynaptic GABAergic SC neurons.The study is very well done. The figures are well laid out and clearly establish the conclusions. My main comments are related to the comparison to other circuits and further questions that might be addressed in the SC.It is stated several times that the superior colliculus and the visual cortex are the two major brain areas for visual processing and these areas are compared throughout the manuscript. However, since both the dorsal lateral geniculate nucleus (dLGN) and SC include similar synaptic motifs, including triadic arrangements of retinal boutons with GABAergic and nonGABAergic neurons, it might be more relevant to compare and contrast retinal convergence and other features in these structures.

Thank you for pointing out that crucial point. Indeed, the comparison to the thalamus is a valid argument, as both the SC and LGN are primary targets of RGC axon terminals. During the preparation of the manuscript, we extensively discussed whether to compare our new SC dataset with existing literature on the LGN or the primary visual cortex (V1) is the more appropriate. Ultimately, we decided on using the visual cortex as the main comparison because of the following reasons:

1. The SC is widely recognized as an evolutionary conserved circuit for visual computation and visually guided behaviors, while the dLGN is generally regarded as a relay station for RGC information to the visual cortex (Steriade, McCormick, 1997). Thus, we believe it is more relevant to compare the evolutionary older visual circuit (SC) to the evolutionary newer visual circuit (visual cortex).

2. In the mouse, the dLGN contains only a limited number of inhibitory interneurons and represent only approximately 6% of the total dLGN neuronal population (Butler, 2008; Evangelio et al., 2018). It has been suggested that the rodent somatosensory thalamus even lacks interneurons (Arcelli et al., 1997). Consequently, directly comparing inhibitory interneurons in the SC to those in the dLGN would pose challenges.

3. Along the same line, the density and also the diversity of inhibitory neurons in the SC is high and likely more comparable to the density and diversity of inhibitory neurons in the visual cortex, than to the dLGN circuit. In the dLGN, TC projection neurons far outnumber inhibitory neurons (Arcelli et al., 1997; Evangelio et al., 2018) and the dLGN is inhabited by just 1-2 classes of GABAergic retinorecipient interneurons (Arcelli et al., 1997; Jaubert-Miazza et al., 2005; Krahe et al., 2011; Ling et al., 2012). Classification approaches (e.g. 3D reconstruction) so far have not revealed any subclasses except for distinctions in intrinsic membrane properties (Leist et al., 2016), suggesting low interneuron diversity in the dLGN. This is in contrast to the vLGN, where a recent study found a diversity of GABAergic neurons (Sabbagh et al., 2021).

4. In the thalamo-cortical circuit, there exists a notable difference in how cortical excitatory and cortical inhibitory neurons are driven by their thalamic input (Alonso and Swadlow, 2005; Cruikshank et al., 2007). This discrepancy forms the basis for several models of visual processing in the visual cortex (Kremkow et al., 2016; Taylor et al., 2021). Which is why we wanted to assess whether the SC follows similar or different rules.

That said, the reviewer is correct that the dLGN and the SC share certain wiring motifs, such as the triadic arrangements of retinal boutons. Unfortunately, the VGAT-ChR2 mouse line used in our study does not specifically label SC inhibitory neurons that are involved in the formation of triadic arrangements. Therefore, we are unable to draw specific conclusion regarding this point. To further investigate this aspect, the usage of GAD67 mice, which have been shown to selectively label intrinsic interneurons which receive RGC input and contact non-GABAergic dendrites (Whyland et al., 2020), would be necessary. Nonetheless, we acknowledge the question raised by the reviewer and in response, we have now provided a more in-depth comparison to the dLGN in the discussion section of the revised manuscript (line 565).

The GABAergic and nonGABAergic neurons showed a wide range of firing rates. It might be interesting to sort the cells by firing rates to see if they exhibit different properties. For example, since the SC contains both GABAergic interneurons and projection neurons it would be interesting to examine whether GABAergic neurons with higher firing rates exhibit narrower spikes, similar to cortical fast spiking interneurons. Similarly, it might be of interest to sort the neurons by their receptive field sizes since this is associated with different SC neuron types.

We thank the reviewer for the interesting suggestions of SC neurons classification into different categories. The relationship between connectivity measures and RF size has been addressed in Author response image 5. We have now studied the relationship of spike waveforms and several measures such as firing rate and RF size in more detail (Author response image 6).

As the baseline firing is generally low in SC and our experiments are performed under anesthetized conditions, we used the evoked firing rates to sort the cells by firing rates or RF sizes. We have added an analysis showing the mean firing rate (calculated over the full recording duration) as a function of the spike width (peak-to-trough duration). We observe no significant relationship between the different groups of cell types. The same accounts if we sort the SC neurons by their RF size. RF sizes were calculated from PSTHs and summed RF for SL and SD. We do not see a relationship between neuron type and firing or RF size.

**Author response image 6. sa4fig6:** Mean firing rate (left) and RF size (right) as a function of peak-to-trough (PT) duration for excitatory and inhibitory SC neurons. Both measures are not correlated to the PT duration (Pearson correlation coefficient, two-sided Wilcoxon rank-sum test).

The recording techniques allowed for the identification of the distance between connected retinocollicular fibers and postsynaptic neurons. It might also be interesting to compare the properties of connected pairs recorded at dorsal versus ventral locations since neurons with different genetic identities and response properties are located in different dorsal/ventral locations (e.g. Liu et al. Neuron 2023). Also, regarding the strength of connections, previous electron microscopy studies have shown that the retinocollicular terminals differ in density and size in the dorsal/ventral dimension (e.g Carter et al JCN 1991).

We thank the reviewer for raising this interesting and relevant point to compare the properties of the connected pairs across the dorsal and ventral location. Unfortunately, our tangential recording approach is not ideally suited for comparing the properties of neurons across the different SC depths. For comparing dorsal versus ventral located neurons in the SC, as done in Liu et al., Neuron 2023, vertical recordings would be more appropriate. We now provide a discussion on this aspect (line 589).

Was optogenetic activation of GABAergic neurons ever paired with visual activation? It would be interesting to examine the receptive fields of the nonGABAergic neurons before and after activation of the GABAergic neurons (as in Gale and Murphy J Neurosci 2016).

This is an important point and indeed we have paired activation of GABAergic neurons with visual stimulation (checkerboard stimulus) to assess the impact of the GABAergic neurons on the firing of the excitatory neurons. We observed a diversity of effects, with some EXNs being strongly suppressed and others being only weakly suppressed. Thus, we predict that the receptive field of those EXN that are suppressed by optogenetically evoked IN firing, should be affected in some way. However, the checkerboard stimulus was only presented for a short duration (1 s) and for only a few trials (n = 30). Therefore, estimating the receptive fields of EXN before and after optogenetic activation of GABAergic neurons is unfortunately not possible with the existing dataset.We now mention this point in the discussion (line 668).

**Reviewer #3 (Public Review):**
This study performs in vivo recordings of neurons in the mouse superior colliculus and their afferents from the retina, retinal ganglion cells (RGCs). Building on a preparation they previously published, this study adds the use of optogenetic identification of inhibitory neurons (aka optotagging) to compare RGC connectivity to excitatory and inhibitory neurons in SC. Using this approach, the authors characterize connection probability, strength, and response correlation between RGCs and their target neurons in SC, finding several differences from what is observed in the retina-thalamus-visual cortex pathway. As such, this may be a useful dataset for efforts to understand retinocollicular connectivity and computations.Recommendations:
**Reviewer #1 (Recommendations For The Authors):**
Some minor points.Fig.1G shows a difference in mean firing rates between inhibitory and excitatory cells. Please plot the cumulative distribution of firing rates to be able to scrutinize the data better.

We have addressed this issue and updated panel G in Figure 1.

Fig. 2C. The black background color of this plot is black; it is not possible to decipher much, please change it to white

We have now changed panel C in Figure 2 to a white background.

Fig. 4D would be better represented as a histogram since most points overlap.

We now represent panel D in Figure 4 as a histogram.

Citations. I would cite some of the foundational work, in some instances, e.g., in the first sentence (SC receives input from the retina)

We have now addressed this issue and cited more foundational studies (e.g. line 68)

The discussion is a bit long; the last paragraph can be removed, mainly because the previous section conflates superficial SC with the entire SC, which is confusing (e.g., Ayupe et al.). In this way, there is more space to discuss the direct implication of the study within the context of known cell types.

We now shortened the discussion and provide more background about different SC cell types in the discussion (line 643).

**Reviewer #2 (Recommendations For The Authors):**
Minor correction: Whyland et al 2020 did not identify V1 input to horizontal cells. A more appropriate reference is Zingg et al Neuron 2017.

We thank the reviewer for this important point and have now corrected the citation in line 613 in the discussion to Zingg et al 2017.

**Reviewer #3 (Recommendations For The Authors):**
Regarding the degree of convergence from RGC to SC, the Crair lab (Furman 2013) performed a quantal analysis in slice that is worth citing.

We included this citation in the revised version of the manuscript (line 501).

I have lost track at this point, but many labs (Heimel, Meister, Farrow, Cang, Isa, maybe others?) have observed that neighboring SC neurons have similar tuning for direction/orientation, but the circuit mechanisms are not well understood. Given the relatively weak correlation between response tuning of RGC axons and their SC target neurons, a useful comparison might be that of SC neurons and their neighbors, and whether SC neurons that show weaker correlation to their RGC axons show stronger correlations with their SC neighbors, which could implicate local connectivity within SC.

We thank the reviewer for providing this interesting comment. With our recording approach we could study locally connected SC neurons. However, the focus of our study was to first characterize the retinocolliculuar connectivity and therefore investigating the intracollicular connectivity is beyond the scope of the current study. We thank the reviewer for the valuable suggestion and will consider to tackle this aspect in a separate study in the future.

Is it possible any of these measurements are biased by laminar targeting of their probe within superficial SC? Their schematic seems to suggest they targeted the deeper part of superficial SC. Do they know whether they recorded throughout superficial SC or targeted the deeper layers closer to stratum opticum?

Our recordings are in between the deeper and upper visual SC layer depending on the recording site on the Neuropixels probe as we use an angled insertion approach. Besides DiI staining (Author response image 7), we can estimate the location of the probe using functional measurements,i.e. visually driven channels and retinotopic locations of the recording sites. If the Neuropixels probe is inserted too superficial, the number of recording site with visually driven activity is low. If the Neuropixels probe is inserted too deep in the visual layers we see two separated regions on the probe with visually driven activity in which the retinotopy is non-continues (please refer to Figure 2 in Sibille et al., 2022). In the recordings included in this study, the number of visually driven channels was generally high and the retinotopy continues, suggesting that we covered a region within the deeper and upper visual layers.

**Author response image 7. sa4fig7:** Functional estimation of probe location. DiI staining of Neuropixels probe (middle) and multi-unit activity across channels in response to visual stimulation (bottom). The white dashed lines in the middle and bottom panels mark the rough boundaries of the visual SC layers.

In Fig. 4, the authors argue that firing in inhibitory neurons is less correlated with RGC input. Does their metric for contribution of retinal input control for the fact that inhibitory neurons have higher firing rates overall and, e.g., may be more depolarized at rest and likelier to fire spontaneous spikes but no less likely to be driven by retina? Or is the argument that their visual responses are more likely to be driven by V1 or local connections?

We thank the reviewer for bringing up that point. The contribution measure estimates the fraction of SC spikes that were preceded by an RGC spike and it is thus, in theory, independent of the firing rate of the SC neuron. In practice, however, we agree that high firing SC neurons may be more likely to have a lower contribution value simply because a larger fraction of their spikes is not preceded by the activity of the presynaptic RGC. But this is exactly what we aimed at characterizing with this analysis. Where these non-RGC driven SC spikes originate from, whether from a more depolarized state of the neuron or by other sources such as V1 or local connections, we can only speculate about. That said, please note that despite SC INs having higher firing rates, not all of them show low contribution. Likewise, we also see SC neurons with low firing rates and low contribution values (new Supp Fig. 3).

Minor point: The optotagging in the example cell doesn't cause the cell to fire for ~50 ms? That is odd. Typically, cells classified as optotagged fire within 5-10 ms of light onset. Is that a strange example cell or is there something different about the optotagging approach?

Unfortunately, transient LED light onsets and offsets can induce light artifacts on Neuropixels probes (Jun et al., 2017; Steinmetz et al., 2021) and therefore it is challenging to use brief LED pulses for optotagging with Neuropixels probes. To avoid this overlap of artefacts and LED evoked spikes, we opted for a longer stimulus duration of 100 ms to activate VGAT neurons (Bennett et al., 2019; Siegle et al., 2019). Moreover, instead of a square pulse, we used a slow ramping for light onsets and offsets to minimize the magnitude of induced artifacts. In Author response image 8 we present examples of individual activated VGAT neurons responding to a 100 ms blue light pulse.

**Author response image 8. sa4fig8:** Optotagging approach. Example traces of a single stimulation pulse and protocol used for optogenetic stimulation. Evoked activity in response to LED stimulation (100ms, 100 trials) for six example SC IN neurons.

References

Alonso J-M, Swadlow HA. 2005. Thalamocortical specificity and the synthesis of sensory cortical receptive fields. J Neurophysiol 94:26–32. doi:10.1152/jn.01281.2004

Arcelli P, Frassoni C, Regondi MC, De Biasi S, Spreafico R. 1997. GABAergic neurons in mammalian thalamus: a marker of thalamic complexity? Brain Res Bull 42:27–37. doi:10.1016/s0361- 9230(96)00107-4

Bennett C, Gale SD, Garrett ME, Newton ML, Callaway EM, Murphy GJ, Olsen SR. 2019. Higher-Order Thalamic Circuits Channel Parallel Streams of Visual Information in Mice. Neuron 102:477-492.e5. doi:10.1016/j.neuron.2019.02.010

Butler AB. 2008. Evolution of the thalamus: a morphological and functional review. Thalamus & Related Systems 4:35–58. doi:10.1017/S1472928808000356

Cruikshank SJ, Lewis TJ, Connors BW. 2007. Synaptic basis for intense thalamocortical activation of feedforward inhibitory cells in neocortex. Nat Neurosci 10:462–468. doi:10.1038/nn1861

Evangelio M, García-Amado M, Clascá F. 2018. Thalamocortical Projection Neuron and Interneuron Numbers in the Visual Thalamic Nuclei of the Adult C57BL/6 Mouse. Frontiers in Neuroanatomy 12.

Froudarakis E, Berens P, Ecker AS, Cotton RJ, Sinz FH, Yatsenko D, Saggau P, Bethge M, Tolias AS. 2014. Population code in mouse V1 facilitates readout of natural scenes through increased sparseness. Nat Neurosci 17:851–857. doi:10.1038/nn.3707

Jaubert-Miazza L, Green E, Lo F-S, Bui K, Mills J, Guido W. 2005. Structural and functional composition of the developing retinogeniculate pathway in the mouse. Vis Neurosci 22:661–676. doi:10.1017/S0952523805225154

Jun JJ, Steinmetz NA, Siegle JH, Denman DJ, Bauza M, Barbarits B, Lee AK, Anastassiou CA, Andrei A, Aydın Ç, Barbic M, Blanche TJ, Bonin V, Couto J, Dutta B, Gratiy SL, Gutnisky DA, Häusser M, Karsh B, Ledochowitsch P, Lopez CM, Mitelut C, Musa S, Okun M, Pachitariu M, Putzeys J, Rich PD, Rossant C, Sun W, Svoboda K, Carandini M, Harris KD, Koch C, O’Keefe J, Harris TD. 2017. Fully integrated silicon probes for high-density recording of neural activity. Nature 551:232–236. doi:10.1038/nature24636

Krahe TE, El-Danaf RN, Dilger EK, Henderson SC, Guido W. 2011. Morphologically Distinct Classes of Relay Cells Exhibit Regional Preferences in the Dorsal Lateral Geniculate Nucleus of the Mouse. J Neurosci 31:17437–17448. doi:10.1523/JNEUROSCI.4370-11.2011

Kremkow J, Perrinet LU, Monier C, Alonso J-M, Aertsen A, Frégnac Y, Masson GS. 2016. Push-Pull Receptive Field Organization and Synaptic Depression: Mechanisms for Reliably Encoding Naturalistic Stimuli in V1. Frontiers in Neural Circuits 10.

Leist M, Datunashvilli M, Kanyshkova T, Zobeiri M, Aissaoui A, Cerina M, Romanelli MN, Pape H-C, Budde T. 2016. Two types of interneurons in the mouse lateral geniculate nucleus are characterized by different h-current density. Sci Rep 6:24904. doi:10.1038/srep24904

Ling C, Hendrickson ML, Kalil RE. 2012. Morphology, Classification, and Distribution of the Projection Neurons in the Dorsal Lateral Geniculate Nucleus of the Rat. PLOS ONE 7:e49161. doi:10.1371/journal.pone.0049161

Sabbagh U, Govindaiah G, Somaiya RD, Ha RV, Wei JC, Guido W, Fox MA. 2021. Diverse GABAergic neurons organize into subtype-specific sublaminae in the ventral lateral geniculate nucleus. J Neurochem 159:479–497. doi:10.1111/jnc.15101

Sibille J, Gehr C, Teh KL, Kremkow J. 2022. Tangential high-density electrode insertions allow to simultaneously measure neuronal activity across an extended region of the visual field in mouse superior colliculus. J Neurosci Methods 376:109622. doi:10.1016/j.jneumeth.2022.109622

Siegle JH, Jia X, Durand S, Gale S, Bennett C, Graddis N, Heller G, Ramirez TK, Choi H, Luviano JA, Groblewski PA, Ahmed R, Arkhipov A, Bernard A, Billeh YN, Brown D, Buice MA, Cain N, Caldejon S, Casal L, Cho A, Chvilicek M, Cox TC, Dai K, Denman DJ, de Vries SEJ, Dietzman R, Esposito L, Farrell C, Feng D, Galbraith J, Garrett M, Gelfand EC, Hancock N, Harris JA, Howard R, Hu B, Hytnen R, Iyer R, Jessett E, Johnson K, Kato I, Kiggins J, Lambert S, Lecoq J, Ledochowitsch P, Lee JH, Leon A, Li Y, Liang E, Long F, Mace K, Melchior J, Millman D, Mollenkopf T, Nayan C, Ng L, Ngo K, Nguyen T, Nicovich PR, North K, Ocker GK, Ollerenshaw D, Oliver M, Pachitariu M, Perkins J, Reding M, Reid D, Robertson M, Ronellenfitch K, Seid S, Slaughterbeck C, Stoecklin M, Sullivan D, Sutton B, Swapp J, Thompson C, Turner K, Wakeman W, Whitesell JD, Williams D, Williford A, Young R, Zeng H, Naylor S, Phillips JW, Reid RC, Mihalas S, Olsen SR, Koch C. 2019. A survey of spiking activity reveals a functional hierarchy of mouse corticothalamic visual areas (preprint). Neuroscience. doi:10.1101/805010

Steinmetz NA, Aydin C, Lebedeva A, Okun M, Pachitariu M, Bauza M, Beau M, Bhagat J, Böhm C, Broux M, Chen S, Colonell J, Gardner RJ, Karsh B, Kloosterman F, Kostadinov D, Mora-Lopez C, O’Callaghan J, Park J, Putzeys J, Sauerbrei B, van Daal RJJ, Vollan AZ, Wang S, Welkenhuysen M, Ye Z, Dudman JT, Dutta B, Hantman AW, Harris KD, Lee AK, Moser EI, O’Keefe J, Renart A, Svoboda K, Häusser M, Haesler S, Carandini M, Harris TD. 2021. Neuropixels 2.0: A miniaturized high-density probe for stable, long-term brain recordings. Science 372:eabf4588. doi:10.1126/science.abf4588

Taylor MM, Contreras D, Destexhe A, Frégnac Y, Antolik J. 2021. An Anatomically Constrained Model of V1 Simple Cells Predicts the Coexistence of Push–Pull and Broad Inhibition. J Neurosci 41:7797–7812. doi:10.1523/JNEUROSCI.0928-20.2021

Usrey WM, Reppas JB, Reid RC. 1999. Specificity and Strength of Retinogeniculate Connections.Journal of Neurophysiology 82:3527–3540. doi:10.1152/jn.1999.82.6.3527

Usrey WM, Reppas JB, Reid RC. 1998. Paired-spike interactions and synaptic efficacy of retinal inputs to the thalamus. Nature 395:384–387. doi:10.1038/26487

Whyland KL, Slusarczyk AS, Bickford ME. 2020. GABAergic cell types in the superficial layers of the mouse superior colliculus. J Comp Neurol 528:308–320. doi:10.1002/cne.24754